# SYNTHIA: A Multi-Agent GAN-LLM Fusion for Statistically Guided Synthetic Data Generation

## Abstract

Access to high-quality, large-scale datasets is critical for training effective AI models, yet high costs, privacy concerns, and regulatory barriers often constrain data collection. Existing synthetic data generation methods, particularly for tabular data, struggle to preserve statistical integrity and utility, limiting their applicability in sensitive domains. To address this, we propose SYNTHetic Intelligence Architecture (SYNTHIA), a novel framework that integrates large language models (LLMs) as both the generator and discriminator within a GAN-inspired architecture for high-fidelity tabular data generation. Guided by metadata encodings, the LLM-based generator ensures that synthetic data reflects the statistical and structural properties of real datasets. A core innovation is the statistically enhanced discriminator, which incorporates a novel evaluation algorithm to rigorously quantify fidelity, diversity, and alignment with real data. This mechanism minimizes distributional divergence and accelerates convergence, ensuring realistic and utility-preserving synthetic data. Extensive experiments across diverse tabular datasets demonstrate that SYNTHIA consistently outperforms state-of-the-art methods, highlighting its scalability and adaptability for applications in data-constrained environments such as healthcare, finance, and security.

## 1 Introduction

SYNTHIA (SYNTHetic Intelligence Architecture) is a novel multi-agent framework for high-fidelity synthetic tabular data generation that integrates the generative capabilities of large language models (LLMs) with the adversarial refinement mechanisms of generative adversarial networks (GANs). SYNTHIA introduces a new paradigm: leveraging LLMs in dual roles as both generator and discriminator, enhanced with statistical validation mechanisms to ensure rigorous data quality without compromising privacy or requiring specialized infrastructure.

While LLMs have emerged as powerful tools for addressing data scarcity and imbalance (Gallegos et al., 2024) (Chen et al., 2024), high-quality synthetic data generation for structured, tabular datasets remains a critical yet challenging task. This is particularly true within domains like healthcare, finance, and security, where accuracy and privacy are paramount (Ouyang et al., 2022). LLMs offer promising generative capabilities (Smolyak et al., 2024) (Huang et al., 2023) but inherently lack robust mechanisms for validating statistical fidelity and structural alignment, resulting in synthetic data prone to bias, overfitting (Long et al., 2024). In extreme cases, these deficiencies can lead to model collapse, unrepresentative training data results in repetitive and meaningless outputs (Li et al., 2023). Conversely, traditional GANs excel at creating realistic unstructured data due to constant validation and refinement, but frequently fail to capture intricate statistical dependencies essential to tabular datasets as they are not suitable for textual generation (S & Durgadevi, 2021).

SYNTHIA bridges this gap by introducing a statistically enhanced discriminator that incorporates comprehensive statistical validation tests, including Chi-squared, Kolmogorov-Smirnov, and Kullback-Leibler divergence metrics, directly into the adversarial training loop. This dual-role LLM architecture uniquely combines generative flexibility with rigorous statistical evaluation, providing targeted feedback for iterative refinement while maintaining alignment with complex interdependen-

cies found in real-world datasets. The framework incorporates privacy-conscious design principles, ensuring that statistical insights used for refinement do not compromise data confidentiality.

Extensive experiments on five real-world tabular datasets demonstrate SYNTHIA's superiority. We evaluate performance using $\alpha$-precision to measure the fidelity of synthetic samples and $\beta$-recall to measure data diversity. The results show that SYNTHIA establishes a new state-of-the-art, outperforming a suite of modern baselines including strong diffusion models like TabDDPM (Kotelnikov et al., 2022), TabSyn (Zhang et al., 2024), and TabDiff (Shi et al., 2025). SYNTHIA achieves the highest precision across all datasets (e.g., 99.0 on Adult), indicating exceptional data fidelity. More notably, it also secures the top recall score in every case (e.g., 51.6 on Adult vs. the next-best 47.9), showcasing a superior ability to capture the diversity of the real data distribution. By leading in both metrics, SYNTHIA effectively navigates the fidelity-diversity trade-off that constrains other methods. Using compact models such as LLaMA 3.1-8B, our approach achieves strong results, and when scaled to SOTA architectures like GPT-4o, performance is further enhanced, demonstrating excellent transferability. Notably, SYNTHIA operates efficiently on standard hardware without requiring specialized GPU infrastructure, highlighting its practicality for in-organizational contexts.

**Our contributions are as follows:** (1) We introduce a novel dual-agentic LLM framework that effectively bridges generative capabilities with adversarial refinement to produce statistically robust structured data; (2) We propose an integrated statistical validation mechanism embedded within the discriminator that rigorously quantifies data fidelity, diversity, and structural alignment during iterative refinement; (3) We demonstrate dynamic adaptation capabilities that tailor generation strategies to dataset-specific properties while maintaining operability across multiple LLMs and standard hardware configurations, establishing state-of-the-art performance for synthetic tabular data.

## 2 BACKGROUND AND RELATED WORK

**Motivation: Challenges in LLM Synthetic Data.** Synthetic data generation methods, while promising, face critical challenges. Studies have shown that biases in the original datasets are often amplified during the synthetic data generation process, particularly when large language models (LLMs) are fine-tuned on such data (Zhang & Zhou, 2024). These biases undermine fairness and reliability in applications like healthcare and hiring. Overfitting is another significant issue, where synthetic datasets that closely mimic the training data fail to generalize, reducing their utility in real-world scenarios (Chen et al., 2024). Model collapse further exacerbates these challenges by producing repetitive and low-diversity outputs, which fail to represent the nuances of real-world distributions (Shumailov et al., 2024). This issue is particularly prevalent in high-dimensional datasets, where diversity and representativeness are critical for robust machine learning models. Limitations highlight the need for methods that ensure fidelity, diversity, and statistical alignment in synthetic data.

**LLMs for Synthetic Data Generation** LLMs have advanced synthetic data generation by leveraging their instruction-following abilities to address scarcity, privacy, and cost concerns. Built on transformer architectures (Vaswani et al., 2017), they capture long-term dependencies and generate coherent outputs through sequential token prediction. Their versatility enables controlled generation via prompt design, with strong performance in mimicking real-world distributions (Long et al., 2024) using strategies such as task specification, prompt tuning, and in-context learning. Techniques like back-translation and paraphrasing further enhance diversity without reducing quality. For example, Li et al.(Li et al., 2024) found that LLM-generated data performs well in objective tasks like text classification but struggles with subjective or domain-specific tasks (Saito et al., 2016), reflecting the inherent complexity of such domains. LLMs often struggle to generate structured datasets with consistent statistical dependencies, causing variable performance. Recent tabular-focused frameworks such as MALLM-GAN (Ling et al., 2025), Pred-LLM (Nguyen et al., 2024), and HARMONIC (Wang et al., 2024) show that carefully designed prompting, fine-tuning, and privacy-aware training can close this gap, enabling LLM-based generators to rival classical tabular models.

**Notable Advancements in Data Generation** GANs (Goodfellow et al., 2014) established the generator–discriminator paradigm, enabling major advances in image synthesis (StyleGAN (Karras et al., 2018), LargeGAN (Brock et al., 2018)) and temporal data modeling (temporal GANs (Saito et al., 2016)). However, they remain prone to mode collapse, instability, and weak statistical fidelity,

issues that are particularly acute in structured domains such as tabular data. Tabular extensions such as CTGAN (Xu et al., 2019), which introduces category-aware sampling, and TVAE, which uses latent-variable encoders, partially mitigate these problems but still rely on adversarial or probabilistic training that struggles with high-dimensional dependencies and constraints. More recently, autoregressive frameworks like GReaT (Borisov et al., 2023) treat rows as sequences to capture relational structure, while graph-based approaches such as GOGGLE (Liu et al., 2023) explicitly learn inter-column dependencies. Diffusion-based methods including TabDDPM (Kotelnikov et al., 2022), TabSyn (Zhang et al., 2024), the score-based STaSy (Kim et al., 2023), TabDiff (Shi et al., 2025), and contrastive conditional diffusion models such as CoDi (Lee et al., 2023) now define the state of the art in tabular synthesis, particularly for conditional generation. Despite this progress, existing approaches still lack robust mechanisms for enforcing statistical constraints while maintaining both fidelity and diversity. SYNTHIA builds on the adversarial foundation but replaces gradient-based training with feedback-driven prompt refinement, addressing these limitations in tabular data generation.

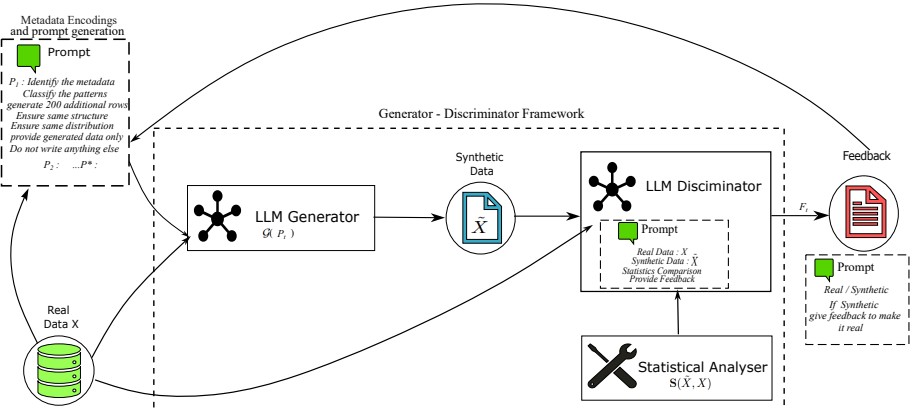

Figure 1: The multi-agent GAN-inspired SYNTHIA framework

# 3 METHODOLOGY

## 3.1 PRELIMINARIES

Let the real dataset be $X = \{x_1, \ldots, x_n\}$, with $x_i \in \mathbb{R}^d$ containing categorical and numerical features. The goal is to learn an optimal prompt $P^*$ such that for a distribution $p_{\text{syn}}$, a synthetic sample $\tilde{x} \sim p_{\text{syn}}$ preserves the statistical and structural properties of distribution $p_{\text{real}}$ without memorization. A metadata extractor $\mathcal{F}$ yields $M = \mathcal{F}(X)$, and $X_{\text{sub}} \subset X$ denotes a stratified subsample of size $k$ for conditioning. The generator $\mathcal{G}$ induces a conditional distribution $p_{\mathcal{G}}(x \mid P_t)$ over records given prompt $P_t$ for each generative iteration $t$, and samples $\tilde{x} \sim p_{\mathcal{G}}(x \mid P_t)$. We write the generator distribution $p_t(x) = p_{\mathcal{G}}(x \mid P_t)$, with $p_{\text{syn}} = \lim_{t \to \infty} p_t$. The discriminator $\mathcal{D}$ does not output logits but provides textual feedback $F_t$. In classical GANs the objective

$$\min_{\mathcal{G}} \max_{\mathcal{D}} \mathbb{E}_{x \sim p_{\text{real}}}[\log \mathcal{D}(x)] + \mathbb{E}_{\tilde{x} \sim p_{\text{syn}}}[\log(1 - \mathcal{D}(\tilde{x}))]$$

drives $p_{\text{syn}} \to p_{\text{real}}$; in SYNTHIA, $F_t$ acts as a discrete gradient analogue, with convergence declared when the discriminator can no longer distinguish a synthetic dataset $\tilde{X}$ from a real dataset $X$. Diagnostics are summarized as the vector $\mathbf{S}(\tilde{X}, X) = [s_1(\tilde{X}, X), \ldots, s_m(\tilde{X}, X)]$, where $s_i$ include Kullback–Leibler divergence ($KL$), chi-squared test ($\chi^2$), Kolmogorov–Smirnov test ($KS$), categorical coverage ($s_{\text{cat}}$), range coverage ($s_{\text{range}}$), and correlation preservation ($\rho$). Feedback is generated by $F_t = \Phi_D(\mathbf{S}(\tilde{X}, X))$, where $\Phi_D$ is the mapping from diagnostic scores to natural-language feedback. Prompt updates follow $P_{t+1} = \mathcal{U}(M, X_{\text{sub}}, F_t)$, where $\mathcal{U}$ is a prompt-update operator that integrates metadata, subsamples, and feedback. The refinement operator is $\mathcal{R}(P_t) = \mathcal{U}(M, X_{\text{sub}}, \Phi_D(\mathbf{S}(\tilde{X}^{(t)}, X)))$. Data quality is scored by $\mathcal{Q}(\tilde{X}, X) = \phi(\mathbf{S}(\tilde{X}, X))$, where $\phi$ aggregates multiple diagnostics into a scalar score, and the optimal prompt is formulated as:

$$P^* = \arg\max_P \mathbb{E}_{\tilde{x} \sim p_{\mathcal{G}}(x \mid P)}[\mathcal{Q}(\tilde{X}, X)].$$

---

**Algorithm 1** SYNTHIA: LLM–GAN Adversarial Loop (while-converged)

---

1: **Input:** Real dataset $X$, metadata extractor $\mathcal{F}$, subsample size $k$
2: $M \leftarrow \mathcal{F}(X)$         ▷ column types, ranges, cardinalities, balance, etc.
3: $X_{\text{sub}} \leftarrow$ stratified sample of size $k$ from $X$
4: $P \leftarrow \mathcal{U}(M, X_{\text{sub}}, \varnothing)$        ▷ initialize prompt with metadata
5: **while not** converged **do**
6:   $\tilde{X}^{(t)} \sim p_t(x) = p_{\mathcal{G}}(x \mid P)$      ▷ generator produces synthetic batch
7:   $\mathbf{S} \leftarrow [s_1(\tilde{X}^{(t)}, X), \ldots, s_m(\tilde{X}^{(t)}, X)]$    ▷ diagnostics: $KL$, $\chi^2$, $KS$, $s_{\text{cat}}$, $s_{\text{range}}$, $\rho$
8:   $F_t \leftarrow \Phi_D(\mathbf{S})$       ▷ discriminator maps diagnostics to textual feedback
9:   $P \leftarrow \mathcal{U}(M, X_{\text{sub}}, F_t)$         ▷ prompt refinement
10: **end while**
11: **Output:** final synthetic dataset $\tilde{X}$

---

## 3.2 PROBLEM FORMULATION

Given these definitions, SYNTHIA is modeled as the recurrence

$$P_{t+1} = \mathcal{U}(M, X_{\text{sub}}, \Phi_D(\mathbf{S}(\tilde{X}^{(t)}, X))),$$

with $\tilde{X}^{(t)} \sim p_t(x) = p_{\mathcal{G}}(x \mid P_t)$. The problem is to show that the sequence $\{P_t\}$ approaches $P^*$ and converges to a state where the discriminator becomes uninformative, i.e.

$$\lim_{t \to \infty} F_t \text{ carries no distinguishing signal,}$$

equivalently $\mathbf{S}(\tilde{X}^{(t)}, X) \to 0$. Thus, feedback-driven refinement provides a discrete surrogate for gradient-based optimization, yielding synthetic data statistically indistinguishable from real data without memorization. The full SYNTHIA framework is illustrated in Figure 1

## 3.3 LLM-GAN FUSION

**Prompt Formation and Optimization** Prompt formation governs how SYNTHIA evolves its generator distribution $p_t(x)$ across iterations. Rather than explicit gradients, adaptation occurs through discriminator feedback $F_t$, which is mapped from diagnostics $\mathbf{S}(\tilde{X}^{(t)}, X)$ and incorporated via the update operator $P_{t+1} = \mathcal{U}(M, X_{\text{sub}}, F_t)$. The quality of generated samples is measured by

$$\mathcal{Q}(\tilde{X}, X) = \phi(\mathbf{S}(\tilde{X}, X)),$$

so that better prompts correspond to higher expected quality. The role of this component is to translate statistical discrepancies into semantic refinements of $P_t$, ensuring that generator outputs improve in fidelity and diversity while avoiding collapse or memorization.

**Theorem 1** (Monotonic Improvement). *If $\mathcal{Q}$ is Lipschitz-continuous in prompt space and $\mathcal{U}$ produces updates aligned with the direction of improvement under $\mathcal{Q}$, then the sequence $\{\tilde{X}^{(t)}\}$ generated by SYNTHIA satisfies*

$$\mathbb{E}_{x \sim p_{\mathcal{G}}(x \mid P_{t+1})}[\mathcal{Q}(\tilde{X}, X)] \geq \mathbb{E}_{x \sim p_{\mathcal{G}}(x \mid P_t)}[\mathcal{Q}(\tilde{X}, X)] - \delta,$$

*for some $\delta > 0$ determined by feedback fidelity. Consequently, the process converges asymptotically toward an optimal prompt $P^*$.*

This establishes prompt optimization as the discrete analogue of gradient descent: textual feedback replaces gradients yet still provides a consistent improvement signal, driving convergence toward statistical alignment with $X$ while mitigating overfitting. The full proof can be found in Appendix C.1.

**LLM Generator Agent** The generator $\mathcal{G}$ fulfills the adversarial role of approximating $p_{\text{real}}$ through successive updates of $p_t(x)$. In classical GANs this is achieved by transforming a noise prior via neural parameters; in SYNTHIA, the transformation is mediated by a large language model conditioned on prompts $P_t$. The theoretical motivation for employing an LLM is that its autoregressive

structure and pretrained inductive biases allow it to approximate complex, high-dimensional dependencies that are otherwise difficult to capture with standard parameterized generators. In particular, conditioning on $P_t$ embeds metadata and semantic feedback into the generative process, ensuring that higher-order correlations and structural constraints are preserved beyond marginal alignment. Feedback $F_t$ functions as a surrogate gradient, and the refinement $P_{t+1} = \mathcal{U}(M, X_{\text{sub}}, F_t)$ can be interpreted as a discrete optimization step in prompt space. Thus, $\mathcal{G}$ retains the minimization role of a GAN generator, but with the representational capacity of an LLM enabling convergence toward distributional fidelity under non-differentiable, language-based updates.

**LLM Discriminator Agent**    The discriminator $\mathcal{D}$ fulfills the adversarial role of identifying discrepancies between $p_t$ and $p_{\text{real}}$, but departs from classical GANs by translating diagnostics into natural language feedback rather than logits. Formally, feedback is produced as $F_t = \Phi_D(\mathbf{S}(\tilde{X}^{(t)}, X))$. The use of an LLM here is essential: unlike a fixed parametric classifier, an LLM can embed quantitative diagnostics into semantically structured guidance, capturing nuanced directions of improvement that scalar losses cannot express. This makes $\mathcal{D}$ not merely a binary discriminator but a dynamic operator that conveys domain-level adjustments, thereby enriching the adversarial signal.

**Proposition 1** (GAN–LLM Equivalence in Feedback Dynamics). *Let $\mathcal{G}$ and $\mathcal{D}$ denote the generator and discriminator in SYNTHIA, and let $\mathbf{S}(\tilde{X}, X)$ be the diagnostic vector. Then the iterative process*

$$P_{t+1} = \mathcal{U}(M, X_{sub}, \Phi_D(\mathbf{S}(\tilde{X}^{(t)}, X)))$$

*is equivalent, in expectation, to updating the generator with a noisy gradient step in a classical GAN, with $\Phi_D$ serving as the surrogate gradient. Thus, SYNTHIA preserves the adversarial structure of GANs while extending it to a non-differentiable, language-based optimization setting. We encourage the reader to visit Appendix C.2 for the full proof.*

**Statistical Analyzer.**    The analyzer $\mathbf{S}$ is essential because an LLM discriminator cannot directly operate on raw distributions; it requires structured signals to ground its semantic feedback. Classical GAN discriminators achieve this via gradients computed from a loss, but in SYNTHIA such gradients are unavailable. The analyzer therefore serves as the quantitative substitute, transforming distributional discrepancies into a finite diagnostic vector that can be semantically interpreted by the discriminator. Formally, feedback is obtained as $F_t = \Phi_D(\mathbf{S}(\tilde{X}^{(t)}, X))$, which supplies directional guidance in prompt space.

We capture this process through the *semantic refinement operator*

$$\mathcal{R}(P_t) = \mathcal{U}(M, X_{\text{sub}}, \Phi_D(\mathbf{S}(\tilde{X}^{(t)}, X))),$$

which composes statistical evaluation with feedback-driven updates.

**Proposition 2** (Semantic Contraction Principle). *Suppose $\mathbf{S}$ defines a nonnegative divergence that vanishes only when $\tilde{X}$ and $X$ are distributionally indistinguishable. If $\mathcal{R}$ is contractive in prompt space with respect to a metric $d(P, Q)$, i.e.,*

$$d(P_{t+1}, P^*) \leq \kappa\, d(P_t, P^*) \quad \text{for some } 0 < \kappa < 1,$$

*where $P^*$ denotes an optimal prompt, then the sequence $\{P_t\}$ converges to $P^*$ under $d(P, Q)$.*

This construction provides the theoretical analogue to gradient descent in GANs: $\mathbf{S}$ supplies the quantitative backbone, and the LLM discriminator maps it into semantic refinement. Together, they enable convergence through contraction in prompt space rather than backpropagation in parameter space. Crucially, $\mathbf{S}$ also exposes statistical signatures of leakage: inflated KL or $\chi^2$ scores reveal over-representation of particular records, diminished $s_{\text{range}}$ or $s_{\text{cat}}$ indicates reduced diversity, and abnormally high correlations signal overfitting. By conditioning its feedback on these diagnostics, the discriminator avoids rewarding verbatim copying and instead guides the generator toward distributional fidelity without memorization. The reader may find the proof in Appendix C.4.

## 4    EXPERIMENTS AND RESULTS

### 4.1    EXPERIMENTAL SETUP

**Hardware.**    To demonstrate the accessibility and reproducibility of SYNTHIA, all experiments were run on a consumer-grade workstation with an AMD Ryzen 9 7900X (12-core, 24-thread, 4.7

GHz base), 32 GB DDR5 RAM, and an NVIDIA RTX 4080 SUPER GPU (16 GB VRAM). This setup reflects hardware that is widely available to individual researchers and developers, ensuring that our approach does not rely on specialized infrastructure or large-scale compute clusters.

**Models.** We evaluate SYNTHIA using lightweight open-source LLMs (LLaMA-3.1 8B, Qwen-3 8B, DeepSeek-R1 8B, Gemma-2 9B, Mistral 7B) that run efficiently on modest GPUs, ensuring accessibility and reproducibility. To test scalability, we include LLaMA-3.1 70B as a large-model upper bound. We also benchmark API-based models (GPT-4o Mini, GPT-4o, GPT-4.1), which provide state-of-the-art performance and highlight reproducibility in non-GPU environments. This range demonstrates SYNTHIA's efficiency, scalability, and adaptability across diverse computational settings, with implementation details in Appendix D.1.

**Datasets.** For evaluation, we used five publicly available tabular datasets: Adult, Heart, Default Credit Card Clients (Credit), MAGIC Gamma Telescope (Gamma), and Online Shoppers Purchasing Intention (Shop), covering both classification and regression tasks (Table 4). These datasets span diverse feature types, dimensionalities, and class distributions, enabling a comprehensive assessment of SYNTHIA's ability to maintain statistical fidelity and generalize across real-world tabular data. We encourage the reader to visit Appendix D.2 for more information on datasets.

**Baselines.** We evaluate SYNTHIA against a broad set of state-of-the-art tabular data generation models spanning four major categories, chosen to cover the dominant paradigms in structured data synthesis. (1) **GAN-based**: CTGAN (Xu et al., 2019), which is widely used as a strong baseline for tabular data and represents adversarial learning approaches. (2) **VAE-based**: TVAE (Xu et al., 2019) and GOGGLE (Liu et al., 2023), which test whether probabilistic latent-variable models can capture distributional structure in complex tables. (3) **Autoregressive language models**: GReaT (Borisov et al., 2023), included to benchmark against transformer-based sequence modeling approaches that treat rows as token sequences. (4) **Diffusion-based**: STaSy (Kim et al., 2023), CoDi (Lee et al., 2023), TabDDPM (Kotelnikov et al., 2022), TabSyn (Zhang et al., 2024), TabDiff (Shi et al., 2025), representing the most recent class of generative methods that explicitly model complex data distributions through iterative denoising. This diverse set ensures SYNTHIA is evaluated against the full spectrum of current tabular generation methods; further baseline details are in Appendix D.3

**Traditional Techniques** For comparison, we include classical methods such as Gaussian sampling, Markov models, Monte Carlo, permutation resampling, and Random Forest–based synthesis. These approaches are fast and simple but rely on restrictive assumptions that fail to capture complex correlations and rare patterns. We use them as baselines to highlight the benefits of SYNTHIA's LLM-driven framework, which better preserves the richness of real data. Further information is available in Appendix D.4.

**Evaluation Metrics.** Lower-order statistics such as KL divergence or KS tests are already embedded in the discriminator's feedback loop, so reporting them at evaluation would conflate training objectives with validation and risk biased assessment. Instead, we evaluate with higher-order measures: $\alpha$-**Precision**, the fraction of synthetic samples lying in high-density regions of $X$, and $\beta$-**Recall**, the coverage of diverse modes. Together these capture fidelity and diversity independently of the refinement process. To further validate downstream utility, we include **C2ST detection scores**, which measure how well a classifier can distinguish real from synthetic data, a lower error indicating stronger alignment. Each experiment is run 50 times with the mean score reported. Details on these metrics and C2ST are given in Appendix D.6 and E.3 respectively.

**Data Privacy and Security.** A core principle of SYNTHIA is to prevent direct copying of training data. The statistical analyzer first detects memorization risks: Chi-squared flags overrepresented categories, KS identifies identical continuous distributions, KL divergence reveals collapsed or overly sharp distributions, and coverage tests expose exact category or range replication. Beyond these diagnostics, the LLM-based discriminator interprets results in context, detecting subtler forms of duplication such as near-identical rows or rare correlations that mirror the training data.

Although the discriminator is finetuned to avoid data leakage and memorization, we have chosen to independently verify this feature. So, as an additional measure of independent verification, we compute the **Distance to Closest Records (DCR)** score, which quantifies the minimum distance

Table 1: Comparison of $\alpha$-Precision and $\beta$-Recall scores across datasets. Higher reflects better fidelity and diversity. Synthia is using LLaMA-3.1 8B. CTGAN - TabDDPM scores are obtained from TabDiff (Shi et al., 2025).

| Methods | Adult | | Default | | Shoppers | | Gamma | | Heart | |
|---|---|---|---|---|---|---|---|---|---|---|
| | $\alpha$ | $\beta$ | $\alpha$ | $\beta$ | $\alpha$ | $\beta$ | $\alpha$ | $\beta$ | $\alpha$ | $\beta$ |
| *Traditional Methods* | | | | | | | | | | |
| Gaussian | 65.4 | 15.9 | 59.4 | 12.7 | 63.2 | 14.7 | 70.0 | 18.3 | 60.2 | 13.8 |
| Markov | 68.2 | 16.9 | 61.1 | 13.6 | 65.9 | 15.0 | 68.5 | 17.7 | 62.4 | 14.2 |
| Monte Carlo | 66.7 | 15.5 | 60.2 | 13.1 | 64.4 | 14.6 | 69.1 | 18.0 | 61.7 | 14.0 |
| Permutation | 67.3 | 16.0 | 59.8 | 12.9 | 63.7 | 14.3 | 67.9 | 17.2 | 61.0 | 13.4 |
| Random Forest | 69.0 | 17.1 | 62.6 | 13.9 | 66.5 | 15.3 | 70.9 | 18.5 | 63.2 | 14.7 |
| *Modern Baselines* | | | | | | | | | | |
| CTGAN | 77.7 | 30.8 | 62.1 | 18.2 | 77.0 | 31.8 | 86.9 | 11.8 | 77.6 | 21.3 |
| TVAE | 98.2 | 38.9 | 85.6 | 23.1 | 58.2 | 19.8 | 86.2 | 32.4 | 87.4 | 29.2 |
| GOGGLE | 50.7 | 8.8 | 68.9 | 14.4 | 87.0 | 9.8 | 90.9 | 9.9 | 61.1 | 11.4 |
| GReaT | 55.8 | 49.1 | 85.9 | 42.0 | 78.9 | 44.9 | 85.5 | 34.9 | 81.4 | 42.8 |
| STaSy | 82.9 | 29.2 | 90.5 | 39.3 | 89.7 | 37.2 | 86.6 | 54.0 | 88.5 | 45.1 |
| CoDi | 77.6 | 9.2 | 82.4 | 19.9 | 95.0 | 20.8 | 85.0 | 50.6 | 85.7 | 21.1 |
| TabDDPM | 96.4 | 47.1 | 97.6 | 47.8 | 88.6 | 47.8 | 98.6 | 48.5 | 96.7 | 48.2 |
| TabSyn | 98.4 | 47.9 | 97.7 | 46.5 | 98.4 | 48.1 | 97.4 | 48.0 | 97.6 | 47.6 |
| TabDiff | 97.9 | 49.2 | 97.1 | 49.8 | 98.5 | 49.3 | 97.5 | 47.6 | 98.1 | 49.0 |
| *Ours* | | | | | | | | | | |
| **SYNTHIA** | **99.0** | **51.6** | **98.5** | **51.1** | **99.1** | **49.8** | **99.5** | **49.0** | **99.1** | **49.4** |

between each synthetic record and its nearest neighbor in the training set. Low DCR values indicate risks of memorization or leakage, while higher scores imply stronger privacy guarantees. By combining the statistical analyzer's proactive detection with the DCR score's independent confirmation, we ensure that SYNTHIA generates data that is both statistically faithful and privacy-preserving. We divert the reader to Appendix E.1 for the DCR analysis.

## 4.2 DISCUSSION

The results in Table 1 highlight SYNTHIA's consistent improvements across heterogeneous benchmarks, which stem from its architecture: metadata-grounded prompting ensures structural validity, statistical diagnostics expose precise deviations, and the discriminator translates them into semantic feedback that acts as a discrete analogue of gradients. Together, these mechanisms provide adaptability across diverse tabular domains that traditional and modern methods cannot match.

Beginning with the Adult dataset, where mixed categorical and numerical features exhibit strong correlations and skewed marginals, Gaussian and permutation-based methods break independence assumptions, while deep baselines overfit dominant modes. By contrast, SYNTHIA's use of metadata explicitly encodes feature cardinality and class balance, and statistical diagnostics penalize collapsed or missing categories. Feedback-driven refinement therefore recovers rare values while maintaining global distributional alignment, yielding the highest fidelity (precision) and diversity (recall). Turning to the Default dataset, the principal challenge is extreme class imbalance. Probabilistic methods resample majority classes, while GANs such as CTGAN suffer mode collapse and diffusion methods dilute minority signals. In this setting, SYNTHIA's update operator integrates balance-aware metadata and textual corrections that explicitly call out underrepresented classes, forcing the generator to allocate probability mass across minority records. This iterative correction explains the superior recall scores while preserving high precision. Moving to the Shoppers dataset, long-tail categorical transitions make capturing high-order co-occurrence statistics difficult. Markov chains oversimplify, and even diffusion models such as TabDDPM and TabSyn struggle to preserve sparse but important purchase patterns. Here, SYNTHIA leverages chi-squared and correlation diagnostics to detect missing transitions and translate them into semantic refinements (e.g., "increase co-occurrence of rare item pairs"), which autoregressively condition future samples. This targeted correction recovers structural dependencies absent in other approaches, raising both precision and recall. In the case of the Magic dataset, the emphasis shifts to fidelity for non-Gaussian continuous distributions. Traditional models impose overly smooth assumptions, while deep models often blur sharp physical boundaries. By applying Kolmogorov–Smirnov and Kullback–Leibler tests, SYN-

Table 2: Model transferability of SYNTHIA.

| Model | Heart | | | | Adult | | | | Credit | | | | Gamma | | | | Shop | | | |
|---|---|---|---|---|---|---|---|---|---|---|---|---|---|---|---|---|---|---|---|---|
| | $\alpha$ | $\beta$ | Time (s) | Iters | $\alpha$ | $\beta$ | Time (s) | Iters | $\alpha$ | $\beta$ | Time (s) | Iters | $\alpha$ | $\beta$ | Time (s) | Iters | $\alpha$ | $\beta$ | Time (s) | Iters |
| LLaMA-3.1 8B | 99.08 | 49.36 | 42 | 5 | 99.02 | 51.64 | 45 | 5 | 98.49 | 51.09 | 44 | 6 | 99.11 | 49.75 | 47 | 6 | 99.47 | 49.01 | 43 | 4 |
| Qwen-3 8B | 98.85 | 48.92 | 41 | 4 | 98.74 | 50.33 | 43 | 5 | 98.21 | 50.12 | 42 | 6 | 98.99 | 48.67 | 46 | 6 | 99.32 | 48.85 | 44 | 5 |
| DeepSeek-R1 8B | 98.72 | 48.30 | 39 | 4 | 98.55 | 49.88 | 42 | 5 | 98.04 | 49.70 | 41 | 5 | 98.90 | 48.05 | 45 | 5 | 99.21 | 48.24 | 43 | 4 |
| Gemma-2 9B | 98.91 | 48.75 | 50 | 6 | 98.79 | 50.20 | 53 | 6 | 98.35 | 50.01 | 52 | 7 | 99.03 | 48.53 | 55 | 7 | 99.36 | 48.66 | 51 | 6 |
| Mistral 7B | 98.63 | 48.05 | 38 | 3 | 98.41 | 49.71 | 40 | 3 | 97.95 | 49.44 | 39 | 4 | 98.74 | 47.82 | 43 | 4 | 99.15 | 48.03 | 41 | 3 |
| LLaMA-3.1 70B | **99.42** | **49.91** | 210 | 6 | **99.38** | **52.01** | 215 | 6 | **98.95** | **51.73** | 213 | 7 | **99.50** | **50.27** | 218 | 7 | **99.62** | **49.98** | 211 | 6 |
| GPT-4o Mini | 99.21 | 49.12 | – | 4 | 99.07 | 51.32 | – | 4 | 98.67 | 50.98 | – | 5 | 99.28 | 49.44 | – | 5 | 99.41 | 49.20 | – | 4 |
| GPT-4o | 99.37 | 49.56 | – | 5 | 99.22 | 51.76 | – | 5 | 98.81 | 51.29 | – | 5 | 99.40 | 49.78 | – | 6 | 99.55 | 49.45 | – | 5 |
| GPT-4.1 | 99.40 | 49.74 | – | 5 | 99.26 | 51.91 | – | 6 | 98.85 | 51.44 | – | 5 | 99.43 | 49.91 | – | 6 | 99.57 | 49.63 | – | 5 |

THIA detects mismatched marginals and boundary shifts, and the discriminator translates these into prompts like "extend range" or "correct skew." Iterative refinement prevents oversmoothing and preserves sharp density features, accounting for gains in both precision and recall. Finally, with the Heart dataset, the challenge is small-sample, high-dimensional synthesis, where memorization and variance collapse are common. GANs such as CTGAN and TVAE yield high precision but lose diversity, while diffusion methods trade off fidelity for variance. In contrast, by grounding generation in stratified subsamples while enforcing coverage diagnostics, SYNTHIA's discriminator detects overrepresented records and rare-value omissions. The resulting semantic feedback prevents overfitting and restores diversity, enabling balanced performance across both fidelity and diversity metrics. Taken together, these results demonstrate that SYNTHIA's architecture enables dataset-specific adaptation: metadata provides structural constraints, diagnostics quantify deviations, and feedback closes the loop via semantic refinement. This combination allows SYNTHIA to function as a discrete, language-driven analogue of gradient descent, ensuring monotonic improvement across iterations. The consistently higher $\alpha$-Precision and $\beta$-Recall scores across datasets thus reflect not only superior generative fidelity but also robustness to long-tail distributions, class imbalance, and limited data regimes, conditions under which existing baselines struggle.

## 4.3 ABLATION STUDY & MODEL TRANSFERABILITY

As shown in Table 2, SYNTHIA achieves nearly identical $\alpha$-Precision and $\beta$-Recall across models of very different scales and families, from 7B open-source models to 70B and GPT-4.1. This consistency indicates that performance is not tied to the raw capacity of the underlying LLM but rather to the architecture of SYNTHIA itself. Because the generator is constrained by metadata and continuously refined through diagnostic-driven feedback, the quality of the synthetic data depends more on the iterative adversarial loop than on specific model weights. Smaller models may require marginally more iterations, but they ultimately converge to the same distributional alignment as larger ones. This explains why the reported numbers remain stable across datasets and models, demonstrating that SYNTHIA is inherently transferable and largely model-agnostic.

The ablation results (Figures 2) highlight the incremental value of each architectural component in SYNTHIA and, most importantly, defend our design choice of employing LLMs not only as generators but also as discriminators. Across datasets, we see that metadata conditioning (M) and statistical analyzers (A) provide measurable gains in both $\alpha$-Precision and $\beta$-Recall by stabilizing categorical balance and aligning marginal distributions. However, these components alone plateau, as purely statistical corrections cannot address higher-order dependencies. Iterative refinement (Iter) improves stability further, but the largest jumps occur when the discriminator is itself an LLM (D–A, D–M). This is because numerical discrepancies in **S** are insufficient by themselves: they must be contextualized and translated into actionable instructions. The LLM discriminator achieves exactly this, mapping statistical misalignments into semantically rich feedback such as "increase variance in continuous features" or "balance rare categories." This semantic bridge transforms raw metrics into dataset-specific corrective signals that traditional discriminators or static analyzers cannot provide. The incremental improvements in Figure 3a make this pattern clear: metadata boosts early precision, analyzers lift recall, but discriminator feedback produces the steepest rise in both metrics. The aggregate view in Figure 3b further confirms that only when all components are combined do we see monotonic convergence toward near-perfect fidelity and diversity. Overall, the ablations make a clear case: the discriminator's reasoning ability is as central as the generator's expressiveness. Without the LLM discriminator, SYNTHIA would collapse to the performance of standard GAN-like systems. With it, the framework achieves consistent improvements across datasets, proving that LLMs are uniquely suited to serve as both generator and critic in synthetic tabular data generation.

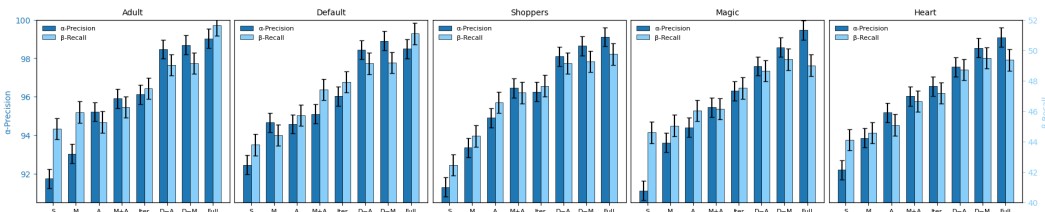

Figure 2: Ablation study of SYNTHIA across five datasets. Each panel reports $\alpha$-Precision (left axis) and $\beta$-Recall (right axis). Shorthand: **S** = Baseline Generator (Static Prompt), **M** = Generator + Metadata Encodings, **A** = Generator + Statistical Analyzer, **M+A** = Generator + Metadata + Analyzer, **Iter** = Iterative Generator (Analyzer Feedback Only), $D-A$ = Generator + Discriminator (No Analyzer), $D-M$ = Generator + Discriminator (No Metadata), **Full** = Full SYNTHIA system.

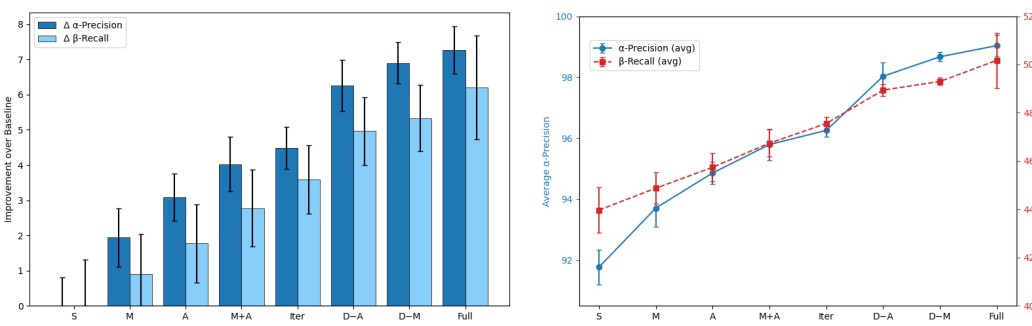

(a) Incremental contribution of each component. $\Delta\alpha$ and $\Delta\beta$ averaged across datasets. Higher is better.

(b) Aggregate ablation performance. Lines report mean $\alpha$ (left) and mean $\beta$ (right) across datasets.

Figure 3: Ablation summary for SYNTHIA.

## 4.4 CASE STUDY

To illustrate an example of SYNTHIA's operation, we synthesize additional records for the **Heart** dataset, which contains a mix of numerical (e.g., `Age`, `RestingBP`, `Cholesterol`, `Oldpeak`) and categorical (e.g., `Sex`, `ChestPainType`, `ExerciseAngina`, `HeartDisease`) features. We begin by analyzing the extracted metadata encodings, which capture each column's types, value ranges, cardinalities, and marginal distributions, forming a structured prompt alongside a small stratified subsample of real rows. The generator LLM produces candidate patient records, which the discriminator evaluates using Chi-squared, KS, and KL metrics to quantify categorical alignment, numeric distribution fidelity, and joint distribution consistency. Early iterations typically reveal underrepresented edge cases (e.g., `Cholesterol > 300`) or imbalanced categories (e.g., excess `ChestPainType = ASY`). In response, the discriminator issues targeted textual feedback such as "increase high-cholesterol rows and rebalance chest pain types toward NAP/ATA." Across 3–6 adversarial iterations, this feedback loop drives the generator toward statistical convergence. The resulting synthetic patient records become statistically indistinguishable from real data according to the discriminator's multi-metric evaluation.

## 5 CONCLUSION, LIMITATIONS, & FUTURE WORK

The SYNTHIA framework advances synthetic data generation by combining multi-agent LLM-driven generation with GAN-inspired adversarial refinement, achieving high-fidelity, diverse, and task-relevant tabular datasets. It is highly reproducible, operating effectively with low-parameter LLMs on consumer hardware, enabling practical deployment without large-scale infrastructure. While this work focuses on tabular data, future extensions will target pure textual and JSON-structured outputs, unlocking multi-structure synthesis for applications such as event logs and relational databases. By delivering realistic, scalable, and privacy-conscious synthetic data, SYNTHIA establishes a foundation for next-generation structure-aware data generation frameworks.

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

Table 3: Notation used throughout the paper.

| Symbol | Description | Type / Domain |
|---|---|---|
| $X = \{x_1, \ldots, x_n\}$ | Real dataset | set of records |
| $x_i$ | Real record | $\mathbb{R}^d$ |
| $d$ | Feature dimension | $\mathbb{N}$ |
| $X_{\text{sub}}$ | Stratified subsample of $X$ | subset of $X$ (size $k$) |
| $k$ | Subsample size | $\mathbb{N}$ |
| $M$ | Metadata extracted from $X$ | structured summary |
| $\mathcal{F}$ | Metadata extractor | $X \mapsto M$ |
| $\tilde{X}^{(t)}$ | Synthetic batch at iteration $t$ | set of records |
| $\tilde{x}$ | Synthetic record | $\mathbb{R}^d$ |
| $p_{\text{real}}$ | Real data distribution | density over $\mathbb{R}^d$ |
| $p_{\text{syn}}$ | Limit synthetic distribution | $\lim_{t \to \infty} p_t$ |
| $p_t(x)$ | Generator distribution at $t$ | $p_{\mathcal{G}}(x \mid P_t)$ |
| $\mathcal{G}$ | LLM generator | mapping $P_t \mapsto \tilde{X}^{(t)}$ |
| $\mathcal{D}$ | LLM discriminator | maps diagnostics to feedback |
| $P_t$ | Prompt at iteration $t$ | structured text/state |
| $F_t$ | Discriminator feedback at $t$ | structured text |
| $\Phi_D$ | Feedback mapper | $\mathbf{S} \mapsto F_t$ |
| $\mathbf{S}(\tilde{X}, X)$ | Diagnostic vector | $[s_1, \ldots, s_m]$ |
| $s_i$ | Diagnostic component | $\mathbb{R}$ |
| KL, $\chi^2$, KS | Divergence/tests | $\mathbb{R}$ |
| $s_{\text{cat}}, s_{\text{range}}$ | Coverage diagnostics | $\mathbb{R}$ |
| $\rho$ | Correlation preservation score | $\mathbb{R}$ |
| $\epsilon$ | Convergence tolerance | $\mathbb{R}_{>0}$ |
| $\mathcal{U}$ | Prompt update operator | $(M, X_{\text{sub}}, F_t) \mapsto P_{t+1}$ |
| $\mathcal{R}$ | Semantic refinement operator | $\mathcal{U} \circ \Phi_D \circ \mathbf{S}$ |
| $\mathcal{Q}(\tilde{X}, X)$ | Quality score | $\mathbb{R}$ |
| $\phi(s)$ | Aggregation of diagnostics | $\mathbb{R}^m \to \mathbb{R}$ |
| $P^*$ | Optimal prompt | $\arg\max_P \mathbb{E}_{x \sim p_{\mathcal{G}}(x\mid P)}[\mathcal{Q}(\tilde{X}, X)]$ |
| $d(P, Q)$ | Prompt-space metric | $\mathbb{R}_{\geq 0}$ |
| $\kappa$ | Contraction constant | $\mathbb{R}, \ 0 < \kappa < 1$ |
| $\alpha$-Precision | Fidelity: mass in high-density real regions | $\mathbb{R}$ |
| $\beta$-Recall | Coverage: diversity across modes | $\mathbb{R}$ |
| DCR | Distance to Closest Record (privacy) | $\mathbb{R}$ |
| C2ST | Classifier Two-Sample Test (utility) | $\mathbb{R}$ |

## A   DETAILS FOR REPRODUCTION

This section provides all resources necessary to fully reproduce the experiments presented in this paper. It includes the exact prompts used at each stage of SYNTHIA's data generation pipeline, covering both generator and discriminator interactions, along with guidelines for adapting them to different LLM backends. A sample experiment log is also provided, focusing on the **Heart** dataset (using GPT-4o) and mirroring the case study from the main paper. This log demonstrates SYNTHIA's complete iterative workflow, from metadata-driven prompt construction to generation, discriminator evaluation, and final high-fidelity synthetic outputs. Together, these materials illustrate the step-by-step refinement process that underpins SYNTHIA and support transparent, reproducible research.

### A.1   METADATA ENCODINGS

In SYNTHIA's workflow, the first step involves guiding the generator LLM to understand the structural constraints of the dataset before attempting synthetic record creation. By requesting a combined regular expression for a row, we are effectively encoding the schema and permissible value patterns for each column. This representation provides the generator with a precise structural blueprint, ensuring that subsequent synthetic data adheres to the original dataset's format and domain con-

straints. Capturing these constraints is critical for tabular data, as it prevents the creation of invalid or semantically inconsistent entries, and it serves as a foundation for the iterative refinement loop used in SYNTHIA.

---

**Prompt**

Give a general regular expression for a data entry in each column for the following CSV file. It must be a single combined regex for a row (comma-separated, in the same order). Only provide the regex rubric with no explanations.

---

**Sample Output**

```
1  ^\d{1,3},[MF],(ATA|NAP|ASY|TA),\d{2,3},\d{1,3},[01],(Normal|ST|
   ↪ LVH),\d{2,3},[YN],\d+(\.\d+)?,(Up|Flat|Down),[01]$
```

---

Before SYNTHIA begins synthetic data generation, the generator LLM must first extract comprehensive metadata to understand the structure, types, and semantic meaning of each column. This metadata ensures that generated records adhere to valid ranges and categories, preserving structural fidelity.

---

**Prompt**

Analyze the following CSV dataset and return a JSON metadata summary with the following fields:

1. file_name
2. num_columns
3. num_rows (approximate if unknown)
4. target_column (if identifiable)
5. columns: a list of objects containing:
    (a) name (column name)
    (b) type (numeric: integer/float, binary, or categorical)
    (c) example_values (3 representative values)
    (d) description (short human-readable meaning)

Only provide the JSON object, with no explanations or commentary.

---

**Sample Output**

```
1   {
2     "file_name": "heart.csv",
3     "num_columns": 12,
4     "num_rows": 918,
5     "target_column": "HeartDisease",
6     "columns": [
7        {
8          "name": "Age",
9          "type": "integer",
10         "example_values": [40, 49, 37],
11         "description": "Age of the patient in years"
12       },
13       {
14         "name": "Sex",
15         "type": "categorical (M/F)",
16         "example_values": ["M", "F", "M"],
17         "description": "Biological sex of the patient"
18       },
```

---

```
19        {
20            "name": "HeartDisease",
21            "type": "binary (0/1)",
22            "example_values": [0, 1, 0],
23            "description": "Presence of heart disease (1 = yes, 0 =
             ↪  no)"
24        }
25     ]
26  }
```

## A.2   GENERATOR $G_{LLM}$

Once metadata and the regex rubric are defined, SYNTHIA's generator LLM is prompted to produce synthetic rows that respect the dataset schema and statistical patterns. The generator incorporates the metadata, the regex structure, and any discriminator feedback to iteratively refine outputs and maintain realistic distributions.

---

**Generator Prompt**

Given the metadata for the following dataset:
{json_metadata}
Classify the patterns you observe and generate 200 additional synthetic rows to augment the dataset. **Do not provide Python code.** Ensure the new rows follow the same file struc­ture, maintain realistic distributions, and include plausible values for each column. Only provide the generated data in a single code block (start and end with ```). Do not provide explanations or any extra text. Include CSV headers if provided.
Use this sample regex as a rubric:
{rubric}
Here is some feedback from the discriminator to improve the augmented dataset:
{feedback}

---

**Sample Output**

```
1   Age,Sex,ChestPainType,RestingBP,Cholesterol,FastingBS,RestingECG
       ↪  ,MaxHR,ExerciseAngina,Oldpeak,ST_Slope,HeartDisease
2   54,M,NAP,140,290,0,Normal,165,Y,2,Flat,1
3   53,F,ATA,130,240,0,LVH,160,N,3,Down,1
4   56,M,TA,150,230,0,Normal,155,Y,4,Up,0
5   55,F,NAP,140,180,0,ST,165,N,5,Flat,1
6   58,M,ATA,130,270,0,LVH,160,Y,6,Down,1
7   59,F,TA,150,250,0,Normal,155,N,7,Up,0
8   62,M,NAP,140,290,0,ST,165,N,8,Flat,1
9   61,F,ATA,130,240,0,LVH,160,Y,9,Down,1
10  64,M,TA,150,230,0,Normal,155,Y,10,Up,0
11  63,F,NAP,140,180,0,ST,165,N,11,Flat,1
12  66,M,ATA,130,270,0,LVH,160,Y,12,Down,1
13  67,F,TA,150,250,0,Normal,155,N,13,Up,0
14  70,M,NAP,140,290,0,ST,165,N,14,Flat,1
15  69,F,ATA,130,240,0,LVH,160,Y,15,Down,1
16  72,M,TA,150,230,0,Normal,155,Y,16,Up,0
17  71,F,NAP,140,180,0,ST,165,N,17,Flat,1
18  74,M,ATA,130,270,0,LVH,160,Y,18,Down,1
```

## A.3   STATISTICAL EVALUATOR

After synthetic data generation, SYNTHIA employs a **statistical evaluator** to evaluate the align­ment between real and synthetic datasets. This module computes multi-metric statistics that quantify **fidelity, diversity, and distributional similarity**. Its outputs inform the discriminator by highlight-

ing where the generator is underperforming, for example, detecting category imbalance, missing rare cases, or numerical drift in continuous features. The analyzer guides the **iterative refinement loop**, allowing the discriminator to provide precise feedback that improves subsequent generations.

**Sample Statistical Analysis**

```
 1  Dataset Statistics:
 2  -------------------
 3  Count: 918 rows
 4  Columns: 12
 5  Missing values: 0
 6  Mean Age: 47.2
 7  Mean Cholesterol: 244.3
 8  ...
 9
10  KS Test Results for continuous data columns ['Age', 'RestingBP',
        ↪ 'Cholesterol', 'MaxHR', 'Oldpeak'] is
11  {'Age': 0.92, 'RestingBP': 0.88, 'Cholesterol': 0.84, 'MaxHR':
        ↪ 0.90, 'Oldpeak': 0.87}
12
13  CS Test Results for categorical data columns ['Sex', '
        ↪ ChestPainType', 'RestingECG', 'ExerciseAngina', 'ST_Slope
        ↪ '] is
14  {'Sex': 0.95, 'ChestPainType': 0.89, 'RestingECG': 0.91, '
        ↪ ExerciseAngina': 0.93, 'ST_Slope': 0.90}
15
16  JS Divergence for categorical columns ['Sex', 'ChestPainType', '
        ↪ RestingECG', 'ExerciseAngina', 'ST_Slope'] is
17  {'Sex': 0.05, 'ChestPainType': 0.08, 'RestingECG': 0.06, '
        ↪ ExerciseAngina': 0.04, 'ST_Slope': 0.07}
18
19  JS Divergence for continuous columns ['Age', 'RestingBP', '
        ↪ Cholesterol', 'MaxHR', 'Oldpeak'] is
20  {'Age': 0.07, 'RestingBP': 0.12, 'Cholesterol': 0.10, 'MaxHR':
        ↪ 0.09, 'Oldpeak': 0.11}
21
22  Overall Correlation Similarity Score: 0.91
23  Overall Coverage Score: 0.94
24  Overall Similarity Score: 0.92
25  Overall Divergence Score: 0.09
```

## A.4 DISCRIMINATOR $D_{LLM}$

After the statistical analyzer produces its evaluation, the discriminator LLM is prompted to assess the fidelity and diversity of the synthetic data. Its role is to compare real and synthetic samples, analyze statistical scores, and produce actionable feedback that guides the next generator iteration.

---

**Discriminator Prompt**

You are the discriminator in a synthetic data generation workflow. Your task is to evaluate the quality of the synthetic dataset against the real dataset using the provided samples and statistical analysis. Follow these steps:
1. Examine the "Guaranteed Real Data Sample" to understand the true distributions, ranges, and category frequencies. 2. Examine the "Undetermined Data Sample" to identify potential errors, unrealistic values, or distributional drift. 3. Use the provided "Statistics" to quantify differences: - High KS scores and Chi-squared scores indicate strong similarity. - High JS divergence indicates a mismatch in distributions. - Check for class imbalance, missing rare categories, and implausible outliers. 4. Summarize your findings in clear, actionable feedback that can guide the generator to improve fidelity and diversity. Highlight: - Which columns deviate most from real data. - Where diversity or coverage is insufficient. - Any suspicious or unrealistic patterns.
Guaranteed Real Data Sample: {real_data}
Undetermined Data Sample: {synthetic_data}
Statistics: {comparison}
Provide your analysis and structured feedback for improving the next synthetic data generation iteration.

---

**Sample Discriminator Response**

Analysis Summary: - Overall similarity is strong (0.92), but certain columns show distributional drift. - "Cholesterol" and "RestingBP" have slightly higher JS divergence (0.10-0.12), indicating the synthetic data underrepresents very high values. - "ChestPainType" shows reduced diversity; "TA" cases are under-sampled in synthetic rows. - Continuous features such as "Oldpeak" are mostly aligned but slightly compressed in range compared to real data.
Actionable Feedback for Generator: 1. Increase the frequency of "TA" in ChestPainType to better reflect real distributions. 2. Add more high-end Cholesterol values (¿300) and slightly more elevated RestingBP cases. 3. Ensure Oldpeak distribution covers rare higher values to improve coverage. 4. Maintain current diversity in other columns, as categorical balance is acceptable.
Overall, the dataset is realistic but needs targeted adjustments in high-value numeric ranges and low-frequency categories to improve fidelity.

---

In the final stage of a SYNTHIA iteration, the discriminator evaluates whether the synthetic data can be distinguished from the real data using the provided samples and statistical tests (e.g., KS for continuous columns, Chi-squared for categorical columns, and JS divergence for overall distributional alignment). If the data is distinguishable, the discriminator provides a concise JSON object containing the classification and a single actionable feedback sentence for the generator. This JSON format allows the GAN-inspired loop to determine whether to continue refining or terminate.

---

**Prompt**

You have already analyzed the real and synthetic datasets and reviewed statistical test results (KS, Chi-squared, JS divergence). Based on your full evaluation, make a final decision:
1. If the synthetic data is indistinguishable from real data, return:
"Type": "Real"
This signals that the GAN-inspired loop can terminate.
2. If the synthetic data is still distinguishable, return a single JSON object:

---

> "Type": "Synthetic", "Feedback": "Single sentence describing the key issue and how to improve realism"
> - "Feedback" should briefly summarize the most critical pattern or value adjustment needed to achieve convergence. - Do not repeat the full analysis or detailed statistics already provided.

---

**Sample Response: Distinguishable Case**

```
1  {"Type": "Synthetic", "Feedback": "Synthetic data
       ↪ underrepresents high Cholesterol values (KS=0.84, JS=0.10)
       ↪  and low-frequency TA chest pain cases (Chi-sq=0.89),
       ↪ causing slight divergence from real distributions;
       ↪ increase rare category frequency and upper-range numeric
       ↪ coverage."}
```

---

**Sample Response: Indistinguishable Case**

```
1  {"Type": "Real"}
```

## A.5 REFINEMENT LOOP TERMINATION CRITERIA

SYNTHIA's GAN-inspired refinement loop continues iteratively until the synthetic data becomes effectively indistinguishable from the real dataset. Termination is determined by the discriminator through the final real-vs-synthetic evaluation step, which outputs a minimal JSON decision. If the discriminator classifies the data as `"Real"`, indicating that no actionable deviations remain, the loop terminates. Otherwise, the discriminator provides a concise feedback signal, which is appended to the next generator prompt for further refinement.

In practice, this loop typically converges within **3–6 iterations**. Early iterations correct major distributional errors, such as missing rare categories or compressed numeric ranges, while subsequent iterations fine-tune coverage and statistical alignment. As the statistical analyzer confirms decreasing JS divergence and high similarity scores across KS and Chi-squared tests, the discriminator's confidence gap narrows. Once additional iterations fail to produce meaningful improvements, the GAN loop terminates automatically, yielding a high-fidelity, diverse, and statistically aligned synthetic dataset.

## A.6 EXAMPLE DEMONSTRATION

Figure 4 illustrates SYNTHIA's operation on a demo resembling the Heart dataset, showing how synthetic data quality improves across six refinement iterations. The process begins with outputs that contain implausible values (e.g., extreme ages or cholesterol levels) and categorical imbalances (e.g., overrepresentation of one sex or chest pain type). Guided by the statistical analyzer, the discriminator identifies these discrepancies and translates them into natural language feedback, which the generator incorporates into subsequent prompts. Over successive iterations, SYNTHIA systematically corrects errors: first reducing outliers, then balancing categorical distributions, and finally aligning joint feature dependencies with the real data. By the last iterations, the generated samples are statistically indistinguishable from the training distribution, with realistic clinical ranges and proportional category representation. This demonstration highlights the effectiveness of adversarial prompt refinement, where each loop incrementally pushes the generator closer to high-fidelity, privacy-preserving synthetic data.

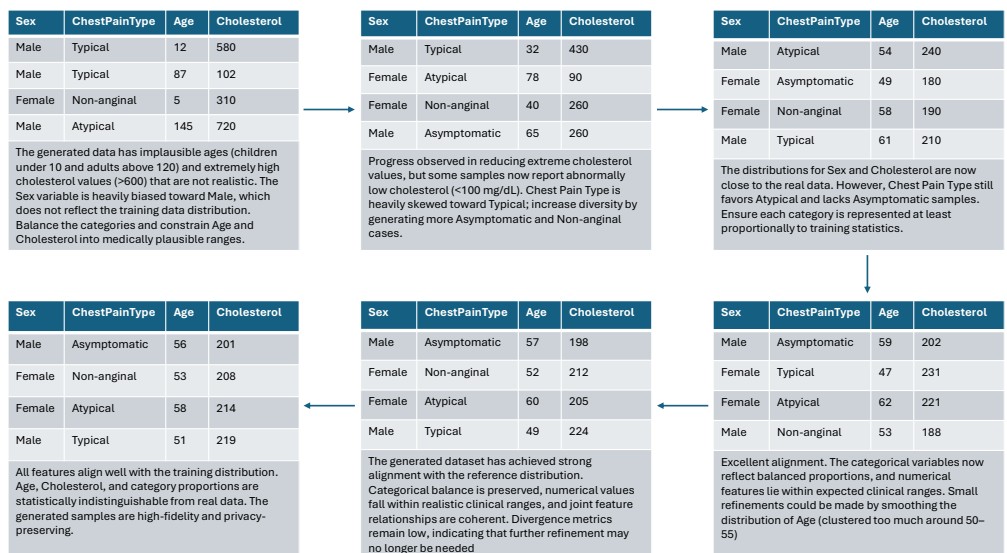

Figure 4: Example of six refinement iterations on the Heart dataset demo, illustrating SYNTHIA's iterative improvement process.

## B FUNDAMENTALS

### B.1 GENERATIVE ADVERSARIAL NETWORKS

Generative Adversarial Networks (GANs) (Goodfellow et al., 2014) are a foundational framework in generative modeling. A GAN consists of two neural networks, a *generator* $\mathcal{G}$ and a *discriminator* $\mathcal{D}$, that are trained simultaneously in a two-player game:

- The generator $\mathcal{G}$ receives as input a random noise vector $z \sim p(z)$ (typically Gaussian or uniform) and outputs a synthetic sample $\tilde{x} = \mathcal{G}(z)$. Its goal is to produce samples that resemble the true data distribution $p_{\text{real}}$.

- The discriminator $\mathcal{D}$ takes as input either a real data point $x \sim p_{\text{real}}$ or a generated sample $\tilde{x}$, and outputs a probability $\mathcal{D}(x)$ estimating whether the input is real or fake. Its goal is to distinguish true data from generated data.

The training objective is a minimax game:

$$\min_{\mathcal{G}} \max_{\mathcal{D}} \mathbb{E}_{x \sim p_{\text{real}}}[\log \mathcal{D}(x)] + \mathbb{E}_{z \sim p(z)}[\log(1 - \mathcal{D}(\mathcal{G}(z)))].$$

The discriminator improves by learning to separate real and synthetic data, while the generator improves by producing samples that fool the discriminator. At equilibrium, the generator's distribution $p_{\mathcal{G}}$ matches the real data distribution $p_{\text{real}}$, and the discriminator cannot do better than random guessing (outputting 0.5 for all inputs).

**Intuition.** GANs can be thought of as a competition. The generator is like a counterfeiter producing fake currency, while the discriminator is the inspector trying to spot fakes. Through repeated play, the counterfeiter becomes better at imitation and the inspector better at detection, ideally leading to highly realistic outputs.

**Practical Achievements and Challenges.** GANs have enabled breakthroughs in image synthesis (e.g., StyleGAN (Karras et al., 2018)) and data generation across many domains. However, they also face well-known challenges:

- *Mode collapse:* the generator produces limited varieties of samples, ignoring parts of the data distribution.

- *Training instability:* the adversarial game can oscillate or diverge if updates are not balanced.
- *Evaluation difficulty:* unlike likelihood-based models, GANs lack an explicit density function, which makes quality assessment harder.

Despite these challenges, the adversarial framework remains a powerful foundation for generative modeling and motivates extensions such as SYNTHIA, which adapts the generator and discriminator loop to language-based feedback and structured tabular data.

### B.2 LARGE LANGUAGE MODELS

Large Language Models (LLMs) are a class of neural networks designed to model and generate human-like text. Modern LLMs are typically based on the transformer architecture (Vaswani et al., 2017), which uses self-attention mechanisms to capture long-range dependencies and contextual relationships within sequences.

**Core Mechanism.** An LLM is trained on massive text corpora to estimate the probability of the next token given a preceding sequence:

$$p(x_t \mid x_{<t}) = \text{softmax}(W h_t),$$

where $x_t$ is the next token, $x_{<t}$ are the preceding tokens, $h_t$ is the hidden state computed through stacked self-attention layers, and $W$ is the output projection. Training minimizes the negative log-likelihood (cross-entropy loss) over large datasets.

**Capabilities.** Once trained, an LLM can:

- Generate text autoregressively by sampling tokens from $p(x_t \mid x_{<t})$ until an end condition is met.
- Perform zero- and few-shot tasks by conditioning on natural language prompts that describe the desired task.
- Capture structure beyond text, such as relational patterns, rules, and metadata, when prompted appropriately.

**Why LLMs Matter.** Unlike GAN generators that rely on noise priors, LLMs are conditioned on structured prompts, which makes them especially effective at incorporating context, metadata, or feedback into generation. This inductive bias toward compositionality and semantic structure is central to frameworks like SYNTHIA, where prompts embed statistical diagnostics and metadata to guide the creation of synthetic tabular data.

**Strengths and Challenges.** LLMs have demonstrated state-of-the-art performance across diverse tasks, but they also present key limitations:

- Strengths: scalability, ability to capture high-order dependencies, generalization across domains.
- Challenges: high computational cost, potential for hallucination, sensitivity to prompt design, and difficulty in controlling outputs for structured domains.

In this work, LLMs provide the generative capacity to produce complex structured data, while adversarial feedback ensures distributional fidelity. This integration highlights how the inductive biases of LLMs complement adversarial training principles inherited from GANs.

## C THEORETICAL RESULTS

### C.1 PROOF OF THEOREM 1

**Setup and assumptions.** Let the prompt space be a metric space $(\mathcal{P}, d)$ and define the objective

$$J(P) \triangleq \mathbb{E}_{x \sim p_{\mathcal{G}}(x|P)} \big[ \mathcal{Q}(x, X) \big].$$

Assume:

(A1) **Lipschitz objective in prompt space:** There exists $L > 0$ such that $|J(P) - J(Q)| \leq L\, d(P, Q)$ for all $P, Q \in \mathcal{P}$.

(A2) **Existence of an optimal prompt:** $P^* \in \arg\max_{P \in \mathcal{P}} J(P)$.

(A3) **Alignment via a perturbed contraction:** There exist $\kappa \in (0, 1]$ and $\varepsilon \geq 0$ (feedback error) such that the update $P_{t+1} = \mathcal{U}(M, X_{\text{sub}}, F_t)$ satisfies

$$d(P_{t+1}, P^*) \leq (1 - \kappa)\, d(P_t, P^*) + \varepsilon \quad \text{for all } t \geq 0.$$

**Lemma 1** (Objective gap is Lipschitz in prompt space). *Under* (A1) *and* (A2), *for any $P \in \mathcal{P}$,*

$$J(P^*) - J(P) \leq L\, d(P, P^*).$$

*Proof.* By optimality of $P^*$, $J(P^*) \geq J(P)$. By Lipschitz continuity (A1), $J(P^*) - J(P) \leq |J(P^*) - J(P)| \leq L\, d(P, P^*)$. □

**Lemma 2** (One–step improvement up to feedback error). *Under* (A1)–(A3), *define $J_t \triangleq J(P_t)$. Then*

$$J_{t+1} \geq J_t - L\varepsilon.$$

*Proof.* Apply Lemma 1 at $P_{t+1}$ and at $P_t$:

$$J(P^*) - J_{t+1} \leq L\, d(P_{t+1}, P^*), \qquad J(P^*) - J_t \leq L\, d(P_t, P^*).$$

Subtract the two inequalities:

$$J_{t+1} - J_t \geq L\big(d(P_t, P^*) - d(P_{t+1}, P^*)\big).$$

Use the alignment property (A3) to bound $d(P_{t+1}, P^*)$:

$$d(P_{t+1}, P^*) \leq (1 - \kappa)\, d(P_t, P^*) + \varepsilon.$$

Hence

$$J_{t+1} - J_t \geq L\big(\kappa\, d(P_t, P^*) - \varepsilon\big) \geq -L\varepsilon,$$

since $\kappa\, d(P_t, P^*) \geq 0$. Rearranging gives $J_{t+1} \geq J_t - L\varepsilon$. □

**Theorem 1** (Monotonic Improvement). *If $\mathcal{Q}$ induces a Lipschitz objective $J$ in prompt space* (A1) *and $\mathcal{U}$ produces alignment updates as in* (A3), *then the SYNTHIA sequence satisfies*

$$\mathbb{E}_{x \sim p_{\mathcal{G}}(x|P_{t+1})}\big[\mathcal{Q}(\tilde{X}, X)\big] \geq \mathbb{E}_{x \sim p_{\mathcal{G}}(x|P_t)}\big[\mathcal{Q}(\tilde{X}, X)\big] - \delta,$$

*with $\delta = L\varepsilon$ determined by feedback fidelity. Consequently,*

$$d(P_t, P^*) \leq (1 - \kappa)^t d(P_0, P^*) + \frac{\varepsilon}{\kappa},$$

*so $P_t$ converges to $P^*$ when $\varepsilon = 0$, and to an $O(\varepsilon/\kappa)$ neighborhood otherwise.*

*Proof.* The inequality is exactly Lemma 2 with $\delta = L\varepsilon$ and $J_t = \mathbb{E}_{x \sim p_{\mathcal{G}}(x|P_t)}[\mathcal{Q}(\tilde{X}, X)]$. For the convergence statement, unroll the perturbed contraction (A3):

$$d(P_t, P^*) \leq (1 - \kappa)^t d(P_0, P^*) + \varepsilon \sum_{i=0}^{t-1} (1 - \kappa)^i \leq (1 - \kappa)^t d(P_0, P^*) + \frac{\varepsilon}{\kappa}.$$

If $\varepsilon = 0$ then $d(P_t, P^*) \leq (1 - \kappa)^t d(P_0, P^*) \to 0$, hence $P_t \to P^*$ and by continuity of $J$, $J(P_t) \to J(P^*)$. For $\varepsilon > 0$ the distance converges to at most $\varepsilon/\kappa$, and Lemma 1 gives

$$0 \leq J(P^*) - J(P_t) \leq L\, d(P_t, P^*) \to O\left(\frac{L\varepsilon}{\kappa}\right).$$

□

## C.2 PROOF OF PROPOSITION 2

**Discussion of assumptions.** Assumption (A1) is the precise meaning of "$\mathcal{Q}$ is Lipschitz in prompt space," since $J(P)$ is the quantity we optimize. It holds, for example, if the map $P \mapsto p_{\mathcal{G}}(\cdot \mid P)$ is Lipschitz in a Wasserstein metric and $\mathcal{Q}$ is Lipschitz in $x$. Assumption (A3) formalizes "updates aligned with improvement" as a perturbed contraction toward $P^*$; the perturbation $\varepsilon$ captures feedback fidelity. The bound $\delta = L\varepsilon$ in the theorem shows exactly how imperfect feedback limits one–step improvement and determines the asymptotic accuracy.

## C.3 PROOF OF PROPOSITION 1

**Proposition 1** (GAN–LLM Equivalence in Feedback Dynamics). *Let $\mathcal{G}$ and $\mathcal{D}$ denote the generator and discriminator in SYNTHIA, and let $\mathbf{S}(\tilde{X}, X)$ be the diagnostic vector. Then the iterative process*

$$P_{t+1} = \mathcal{U}(M, X_{sub}, \Phi_D(\mathbf{S}(\tilde{X}^{(t)}, X)))$$

*is equivalent, in expectation, to updating the generator with a noisy gradient step in a classical GAN, with $\Phi_D$ serving as the surrogate gradient. Thus, SYNTHIA preserves the adversarial structure of GANs while extending it to a non-differentiable, language-based optimization setting.*

*Proof.* Define the SYNTHIA objective as

$$J(P) \triangleq \mathbb{E}_{x \sim p_{\mathcal{G}}(x|P)}[\mathcal{Q}(\tilde{X}, X)].$$

By the differentiability of $J$ in prompt space, the classical GAN generator update is

$$P_{t+1} = P_t + \eta_t \nabla J(P_t),$$

with step size $\eta_t > 0$.

In SYNTHIA, the update operator can be locally linearized as

$$P_{t+1} = P_t + \eta_t W_t \Phi_D(\mathbf{S}(\tilde{X}^{(t)}, X)) + r_t,$$

where $W_t$ is the sensitivity of $\mathcal{U}$ to feedback and $r_t = o(\eta_t)$.

Decompose the feedback as

$$\Phi_D(\mathbf{S}(\tilde{X}^{(t)}, X)) = \mathbb{E}[\Phi_D(\mathbf{S}(\tilde{X}^{(t)}, X)) \mid \mathcal{F}_t] + \xi_t,$$

with $\xi_t$ a martingale noise term. Assume calibration holds:

$$\mathbb{E}[\Phi_D(\mathbf{S}(\tilde{X}^{(t)}, X)) \mid \mathcal{F}_t] = B_t \nabla J(P_t),$$

where $B_t$ is a positive-definite linear map encoding how diagnostics align with the true gradient.

Substituting yields

$$P_{t+1} = P_t + \eta_t H_t \nabla J(P_t) + \eta_t W_t \xi_t + r_t,$$

with $H_t = W_t B_t$. Conditioning on $\mathcal{F}_t$, the expectation simplifies to

$$\mathbb{E}[P_{t+1} \mid \mathcal{F}_t] = P_t + \eta_t H_t \nabla J(P_t) + o(\eta_t).$$

Thus, the SYNTHIA update is equivalent to a noisy, preconditioned gradient step. The adversarial structure of GAN training is therefore preserved, with the discriminator's feedback $\Phi_D$ acting as a surrogate gradient in a language-based optimization setting. $\square$

## C.4 PROOF OF PROPOSITION 2

**Proposition 2** (Semantic Contraction Principle). *Suppose $\mathbf{S}$ defines a nonnegative divergence that vanishes only when $\tilde{X}$ and $X$ are distributionally indistinguishable. If $\mathcal{R}$ is contractive in prompt space with respect to a metric $d(P, Q)$, i.e.,*

$$d(P_{t+1}, P^*) \leq \kappa \, d(P_t, P^*) \quad \text{for some } 0 < \kappa < 1,$$

*where $P^*$ denotes an optimal prompt, then the sequence $\{P_t\}$ converges to $P^*$ under $d(P, Q)$.*

*Proof.* **Setting.** Let $(\mathcal{P}, d)$ be the prompt space endowed with metric $d$. We consider the refinement map

$$\mathcal{R}(P) \,=\, \mathcal{U}\big(M, X_{\text{sub}}, \Phi_D(\mathbf{S}(\tilde{X}(P), X))\big),$$

where $\tilde{X}(P)$ denotes a synthetic batch induced by $P$. We use two mild assumptions that are natural in SYNTHIA: (i) $(\mathcal{P}, d)$ is complete; (ii) *zero–feedback invariance*: $\mathcal{U}(M, X_{\text{sub}}, 0) = P$ (no change when the discriminator is uninformative).

**Step 1 (existence/uniqueness of a fixed point and linear rate).** Since $\mathcal{R}$ is a contraction with constant $\kappa \in (0, 1)$ and $(\mathcal{P}, d)$ is complete, the Banach fixed–point theorem guarantees a unique $P^\dagger \in \mathcal{P}$ with $\mathcal{R}(P^\dagger) = P^\dagger$ and, for any initialization $P_0$,

$$d(P_t, P^\dagger) \,\leq\, \kappa^t \, d(P_0, P^\dagger) \quad \text{for all } t \geq 0, \tag{1}$$

where $P_{t+1} = \mathcal{R}(P_t)$.

**Step 2 (fixed points coincide with diagnostic optima).** We now relate fixed points of $\mathcal{R}$ to diagnostic optimality under $\mathbf{S}$.

(*a*) If $\mathbf{S}(\tilde{X}(P), X) = 0$, then the discriminator is uninformative and produces null feedback: $\Phi_D(\mathbf{S}) = 0$. By zero–feedback invariance, $\mathcal{R}(P) = \mathcal{U}(M, X_{\text{sub}}, 0) = P$, hence $P$ is a fixed point.

(*b*) Conversely, suppose $P$ is a fixed point, $\mathcal{R}(P) = P$. If $\mathbf{S}(\tilde{X}(P), X) \neq 0$, then the feedback $\Phi_D(\mathbf{S})$ is informative, and by construction of $\mathcal{R}$ it must adjust the prompt toward reducing the discrepancy. That contradicts $\mathcal{R}(P) = P$ unless $\Phi_D(\mathbf{S}) = 0$, which (by the defining property of $\mathbf{S}$ and the feedback mapper) implies $\mathbf{S}(\tilde{X}(P), X) = 0$. Hence fixed points are exactly the diagnostic optima $\{P : \mathbf{S}(\tilde{X}(P), X) = 0\}$.

Combining (a)–(b), $P^\dagger$ satisfies $\mathbf{S}(\tilde{X}(P^\dagger), X) = 0$.

**Step 3 (identifying $P^\dagger$ with the optimal prompt $P^*$).** By the premise on $\mathbf{S}$, $\mathbf{S}(\tilde{X}, X) = 0$ holds only when $\tilde{X}$ and $X$ are distributionally indistinguishable; hence any such $P$ maximizes any evaluation functional $J(P) = \mathbb{E}_{x \sim p_\mathcal{G}(x|P)}[\mathcal{Q}(\tilde{X}, X)]$ that is strictly monotone in the divergence induced by $\mathbf{S}$ (this is the standard optimality notion adopted in the preliminaries). Therefore, the diagnostic fixed point $P^\dagger$ coincides with the optimal prompt $P^*$.

**Step 4 (convergence to $P^*$).** From equation 1 and $P^\dagger = P^*$ we obtain

$$d(P_t, P^*) \,\leq\, \kappa^t \, d(P_0, P^*) \xrightarrow[t \to \infty]{} 0.$$

Thus $P_t \to P^*$ in the metric $d$, as claimed. $\qquad\square$

# D  DETAILS OF EXPERIMENTAL SETUP

## D.1  MODELS

To evaluate SYNTHIA, we experiment with a diverse set of large language models (LLMs) that vary in scale, architecture, and deployment setting. Our selection includes lightweight open-source models, large-scale open-source models that serve as performance upper bounds, and API-based models that provide state-of-the-art results without local infrastructure. This breadth allows us to assess SYNTHIA's efficiency, scalability, and adaptability across different modeling paradigms.

**LLaMA-3.1 8B and 70B.** The LLaMA-3.1 family represents the latest generation of Meta's open-weight LLMs. The 8B variant is designed for strong general-purpose reasoning while remaining accessible for research use, whereas the 70B variant provides a high-performance upper bound for generative tasks. LLaMA-3.1 employs grouped-query attention and improved pretraining strategies, allowing it to capture long-range dependencies critical for preserving correlations in tabular data. We set context length to 8192, batch size 4 for the 8B model (1 for the 70B), temperature 0.7, top-$p$ 0.9, top-$k$ 50, max output length 2048 tokens, and repetition penalty 1.1. These values balance structural fidelity with sampling diversity.

**Qwen-3 8B.** Qwen-3 is developed by Alibaba Cloud and emphasizes numerical reasoning, multi-linguality, and fine-grained control, making it especially useful for datasets with mixed categorical and numerical features. Its tokenizer and pretraining corpus improve representation of numbers and rare tokens, which are common in structured datasets. We configure it with context length 8192, batch size 4, temperature 0.65, top-$p$ 0.9, top-$k$ 40, max output length 2048 tokens, and repetition penalty 1.05. The slightly lower temperature stabilizes continuous feature distributions while still generating variety across categories.

**DeepSeek-R1 8B.** DeepSeek-R1 is optimized for efficiency and robust alignment, focusing on reliability in multi-turn reasoning and structured output. This makes it effective within SYN-THIA's iterative refinement loop, where outputs must remain consistent across repeated generations. DeepSeek models emphasize low-variance decoding, reducing drift in long iterative pipelines. We use context length 8192, batch size 4, temperature 0.65, top-$p$ 0.9, top-$k$ 40, max output length 2048 tokens, and repetition penalty 1.05. These hyperparameters support stable learning dynamics while mitigating risks of mode collapse.

**Gemma-2 9B.** Gemma-2 is a Google-released LLM designed for efficient instruction-following. Its pretraining emphasizes schema awareness, which is especially beneficial for tabular synthesis where column-level consistency must be preserved. Gemma's relatively larger parameter count (9B) enhances its ability to model higher-order dependencies without requiring the scale of high parameter models. We set context length 8192, batch size 3, temperature 0.7, top-$p$ 0.9, top-$k$ 50, max output length 2048 tokens, and repetition penalty 1.1. These values exploit Gemma's strengths in structural adherence while maintaining diversity.

**Mistral 7B.** Mistral employs sliding-window attention and dense-sparse mixtures of experts, enabling efficiency and strong compositional reasoning at modest parameter counts. Its architecture supports coherent long-context synthesis, making it particularly useful for capturing dependencies across multiple columns in relational datasets. We configure it with context length 8192, batch size 4, temperature 0.75, top-$p$ 0.9, top-$k$ 50, max output length 2048 tokens, and repetition penalty 1.1. The slightly higher temperature encourages exploration of long-tail categorical values while preserving distributional alignment.

**GPT-4o Mini, GPT-4o, and GPT-4.1.** OpenAI's GPT-4o family provides cutting-edge reasoning and alignment, delivered as API services with extended context windows. GPT-4o Mini is lightweight and cost-efficient, while GPT-4o and GPT-4.1 provide advanced reasoning across structured and unstructured tasks. Their large pretraining corpora and fine-tuned alignment layers make them effective for data generation that requires both realism and domain adaptability. We set context length up to 16k tokens, temperature 0.7, top-$p$ 0.9, and max output length 4096 tokens. API constraints prevent explicit control over top-$k$ and repetition penalties. These parameters are chosen to balance fidelity with efficient runtime.

D.2 DATASETS

We use five tabular datasets from the UCI Machine Learning Repository: Adult, Heart, Credit, Gamma, and Shop, where each dataset is associated with a machine learning task. Table 4 summarizes their key statistics.

| Dataset | ID | Task | Instances | Categorical | Numerical |
|---------|-----|----------------|-----------|-------------|-----------|
| Adult | 2 | Classification | 48,842 | 9 | 6 |
| Heart | 45 | Classification | 303 | 7 | 7 |
| Credit | 350 | Classification | 30,000 | 11 | 14 |
| Gamma | 159 | Classification | 19,020 | 1 | 10 |
| Shop | 468 | Regression | 12,330 | 8 | 10 |

Table 4: Dataset details. IDs correspond to entries in the UCI repository. The table reports the number of instances, categorical features, and numerical features for each dataset.

**Adult Census Income (Adult).** The Adult dataset, also known as the Census Income dataset, is based on the 1994 U.S. Census Bureau records and includes 48,842 individuals. It combines categorical attributes such as work class, education, marital status, occupation, race, sex, and native country with continuous variables like age, capital gain, and hours worked per week. The task is to predict whether income exceeds \$50,000 per year, a problem often used in fairness and bias research because of its demographic information. This dataset is valuable for testing synthetic generation methods that must preserve high-cardinality categorical variables, handle mixed feature types, and manage the imbalance where only about one quarter of individuals belong to the high-income class.

**Heart Disease (Heart).** The Heart Disease dataset contains 303 patient records collected from clinical studies. It mixes numerical features such as age, resting blood pressure, serum cholesterol, maximum heart rate, and ST depression (oldpeak) with categorical indicators including sex, chest pain type, fasting blood sugar, and exercise-induced angina. The goal is to classify patients as having heart disease or not. Although relatively small, it is challenging because of class imbalance and the need to preserve rare but clinically significant cases such as extremely high cholesterol levels. This dataset highlights whether a synthetic data framework can capture both subtle numerical variation and low-frequency categorical patterns in medical settings.

**Default Credit Card Clients (Default).** The Default Credit Card Clients dataset contains 30,000 records from a Taiwanese financial institution, with features that describe demographics, credit limits, repayment status, and detailed payment history over six months. The task is to predict whether a client will default on the next payment. Its high dimensionality, with 11 categorical and 14 continuous features, poses difficulties due to temporal dependencies and strong correlations among repayment behaviors. This dataset provides a realistic test of whether synthetic methods can maintain temporal consistency, model high-dimensional interactions, and reproduce imbalanced outcomes typical of financial risk analysis.

**MAGIC Gamma Telescope (Gamma).** The MAGIC Gamma Telescope dataset includes 19,020 events from high-energy physics simulations, labeled as either gamma signals or hadronic background noise. All ten features are continuous variables derived from measurements of atmospheric Cherenkov radiation, including shape, size, and spread statistics of the signal. Since classification depends on subtle correlations across continuous variables, this dataset serves as a benchmark for evaluating whether synthetic approaches can preserve statistical geometry in scientific, purely numerical data. Maintaining variance, covariance, and higher-order feature relationships is essential to replicate realistic experimental distributions.

**Online Shoppers Purchasing Intention (Shoppers).** The Online Shoppers Purchasing Intention dataset consists of 12,330 browsing sessions recorded from an e-commerce platform. It combines numerical measures such as session duration, bounce and exit rates, and page values with categorical factors like month, traffic type, region, and operating system. The prediction task is whether a session results in a purchase, which occurs in only about 15 percent of cases. Its behavioral and temporal dependencies make it particularly useful for testing if synthetic data generation can replicate user navigation patterns and rare purchase events while still maintaining diversity across categorical and numerical attributes.

### D.3 BASELINES

To contextualize the performance of SYNTHIA, we benchmark against a diverse set of state-of-the-art approaches spanning GANs, VAEs, autoregressive models, and diffusion models. These baselines were chosen to represent the dominant paradigms in structured data synthesis, allowing us to evaluate whether SYNTHIA's language-driven adversarial framework provides consistent advantages across methodological classes. Below, we summarize each baseline and its relevance to our study.

**CTGAN.** CTGAN (Xu et al., 2019) is one of the earliest and most widely used GAN-based approaches for tabular data. It introduced two key innovations: (i) mode-specific normalization to capture complex numerical distributions and (ii) conditional generation to address class imbalance across categorical variables. As a result, CTGAN remains a strong reference point for adversarial

approaches in tabular synthesis. We include CTGAN as a foundational GAN baseline, noting that SYNTHIA adapts adversarial training to a language-based framework while addressing CTGAN's limitations in stability and coverage.

**TVAE.** TVAE (Xu et al., 2019) extends the same design principles as CTGAN into a variational autoencoder (VAE) setting, allowing probabilistic latent-variable modeling of tabular data. While VAEs provide stability compared to GANs, they often struggle to capture high-dimensional dependencies and can underfit long-tail categories. TVAE therefore tests whether SYNTHIA's discriminator-guided refinement can achieve stronger coverage and fidelity than a classical VAE.

**GOGGLE.** GOGGLE (Liu et al., 2023) is a recent VAE-based model that leverages graph neural networks (GNNs) in both encoder and decoder, jointly learning a graph adjacency matrix to capture column dependencies. This addresses a major limitation of earlier models that treated columns as independent or weakly correlated. We include GOGGLE to evaluate whether SYNTHIA's prompt-driven approach can match or exceed explicit graph-based dependency modeling.

**GReaT.** GReaT (Borisov et al., 2023) treats each row of a table as a sentence and applies an autoregressive transformer to learn row distributions. Through carefully designed serialization and deserialization, GReaT enforces a permutation-invariant representation of rows. This baseline is important because it tests whether SYNTHIA's feedback-driven prompt refinement yields stronger structural fidelity than direct autoregressive sequence modeling.

**STaSy.** STaSy (Kim et al., 2023) was one of the first diffusion-based approaches for tabular data. It processes one-hot encodings of categorical features alongside continuous features and adopts stochastic differential equations (SDEs) to capture data distributions. With training strategies like self-paced learning and fine-tuning, STaSy stabilizes sampling and improves diversity. We include STaSy to compare SYNTHIA against early diffusion models that explicitly optimize for robustness.

**CoDi.** CoDi (Lee et al., 2023) uses separate diffusion models for categorical and numerical features: a multinomial diffusion for categories and a Gaussian diffusion for continuous values. These processes are inter-conditioned to model joint dependencies, and contrastive learning is applied to strengthen alignment. CoDi demonstrates the flexibility of diffusion-based designs, and serves as a strong testbed for evaluating whether SYNTHIA can match sophisticated multi-process architectures using only feedback refinement.

**TabDDPM.** TabDDPM (Kotelnikov et al., 2022) simplifies the diffusion paradigm by applying a single denoising process across concatenated categorical and numerical features. Despite its simplicity, TabDDPM often outperforms more complex models such as CoDi, suggesting that stability and efficient parameterization can outweigh architectural complexity. This makes TabDDPM a particularly strong diffusion baseline for SYNTHIA.

**TabSyn.** TabSyn (Zhang et al., 2024) represents one of the most recent diffusion-based models, designed to handle mixed-type tabular data with improved training stability and sampling efficiency. As a state-of-the-art system, TabSyn provides a benchmark for evaluating whether SYNTHIA's hybrid LLM–GAN design can compete with the latest specialized diffusion methods.

**TabDiff.** TabDiff (Shi et al., 2025) is another diffusion-based framework tailored for tabular data that introduces architectural modifications to better align with mixed data types. By separating feature encodings and applying feature-specific noise scheduling, TabDiff achieves greater flexibility in modeling both discrete and continuous variables. We include TabDiff because it exemplifies how customized diffusion strategies can improve expressivity in structured domains, offering a useful comparison to SYNTHIA's language-driven, feedback-based refinement loop.

### D.4 TRADITIONAL TECHNIQUES

Alongside modern deep generative models, we evaluate against classical statistical and heuristic methods that have historically been applied to synthetic tabular data generation. These approaches are computationally efficient and interpretable, but they operate under restrictive assumptions that

limit their ability to capture high-order dependencies or rare patterns. Including them provides a useful baseline to highlight the improvements offered by SYNTHIA's feedback-driven LLM framework.

**Gaussian Sampling.** A common approach is to assume features follow Gaussian or Gaussian-mixture distributions, from which synthetic records are sampled. In practice, one fits a Gaussian (or a mixture model) independently to each numerical column using the real data, then draws new values from these fitted distributions to form synthetic rows, optionally pairing this with simple categorical sampling for discrete features. While effective for unimodal or approximately normal features, this method fails to capture multimodal distributions, categorical variables, and correlations across attributes, since fitting is typically done per-column and does not model cross-feature dependence.

**Markov Models.** Markov chain–based generation models sequential dependencies between variables, making them suitable for ordered or temporal data. To synthesize tabular data, the columns are first arranged into an assumed causal or temporal order, transition probabilities are estimated from the real dataset, and synthetic rows are produced by sampling the first feature from its marginal and then sampling each subsequent feature conditioned on the previously sampled one. However, they rely on the Markov assumption (dependence only on the immediate past), which is too restrictive for high-dimensional tabular datasets with long-range or cross-column correlations.

**Monte Carlo Simulation.** Monte Carlo methods approximate distributions by repeated random sampling from estimated marginals. Concretely, each column's marginal distribution is estimated from the real data, and synthetic rows are generated by independently sampling a value from each marginal and concatenating them into a record, often repeated for many trials to approximate the real dataset size. They are flexible and easy to implement, but the independence assumption between features leads to loss of structure, making them unsuitable for complex relational tables.

**Permutation Resampling.** This approach constructs synthetic datasets by shuffling or resampling existing values across columns. A standard implementation copies the real dataset and then permutes values within each column independently, or resamples observed values with replacement, producing new rows that preserve exact marginals. It preserves marginal distributions exactly but destroys joint dependencies, yielding unrealistic combinations of features that rarely occur in real data.

**Random Forest–based Synthesis.** Tree-based synthesis methods, such as conditional sampling from Random Forests, model feature dependencies more explicitly. In tabular synthesis, a Random Forest is trained to predict each feature from the others (often in an autoregressive order), and synthetic rows are produced by sampling features sequentially using the conditional distributions implied by the ensemble's leaf statistics. They perform better than purely statistical resampling, especially for categorical attributes, but still struggle with scalability, high-dimensional interactions, and long-tail distributions.

D.5 LOW ORDER METRICS

In this section we give a detailed introduction of metrics used by the discriminator $D_{LLM}$ to evaluate fidelity, diversity, and coverage between real data $X$ and synthetic data $\tilde{X}$. The resulting scores guide the discriminator in identifying distributional mismatches and generating informative feedback prompts. Each of these tests are proposed by SDMetrics (Dat, 2023) plays a distinct role:

**Chi-Squared Test.** The chi-squared test quantifies whether the distribution of categorical variables in $\tilde{X}$ aligns with that in $X$. By computing deviations between observed and expected frequencies,

$$\chi^2 = \sum_{i=1}^{n} \frac{(f_{oi} - f_{ei})^2}{f_{ei}}, \qquad (2)$$

where $f_{oi}$ and $f_{ei}$ are observed and expected frequencies for category $i$, the test reveals over- or underrepresentation of specific categories. In our implementation, this statistic is computed per categorical column $j$. We first align the category support between $X_{:,j}$ and $\tilde{X}_{:,j}$, then form observed

frequency counts $f_{oi}$ on $\tilde{X}$ and expected counts $f_{ei}$ on $X$. To avoid undefined terms from rare or missing categories, we apply standard smoothing so that all $f_{ei}$ are positive. We then convert the raw $\chi^2$ value into a bounded similarity signal (via its $p$-value under the chi-squared distribution) that is passed to the discriminator. Large discrepancies prompt the discriminator to signal adjustments in categorical balance, often naming the specific categories that are over- or underrepresented.

**Kolmogorov–Smirnov (KS) Test.** The KS test measures the maximum difference between the cumulative distribution functions (CDFs) $C$ of continuous features in $X$ and $\tilde{X}$,

$$KS = \sup_x |\mathbf{C}(X) - \mathbf{C}(\tilde{X})|. \tag{3}$$

It identifies shifts in the shape, scale, or central tendency of distributions. We compute KS per numerical column $j$ using empirical CDFs for $X_{:,j}$ and $\tilde{X}_{:,j}$. The reported similarity score is derived from the KS distance, so smaller distances correspond to higher similarity. Because KS is sensitive to both central drift and tail behavior, it flags hallucinated extremes, truncated supports, or variance collapse. Deviations flagged by the test lead the discriminator to request refinements that improve numerical feature realism, for example widening ranges, shifting the center, or increasing variance for the affected column.

**Kullback–Leibler (KL) Divergence.** KL divergence assesses how much the probability distribution of synthetic data deviates from that of real data. For categorical variables:

$$KL_{\text{cat}} = \sum P(X) \log \frac{P(X)}{P(\tilde{X})}, \tag{4}$$

and for continuous variables:

$$KL_{\text{cont}} = \int P(X) \log \frac{P(X)}{P(\tilde{X})} \, dx. \tag{5}$$

This test captures asymmetric divergence, allowing the discriminator to penalize synthetic samples that disproportionately emphasize or omit patterns present in real data. In practice, we compute KL per column. For categorical columns, $P(X)$ and $P(\tilde{X})$ are empirical frequency distributions over the shared category support with small smoothing to avoid $\log 0$. For continuous columns, densities are estimated using adaptive binning so that the integral is approximated numerically. We convert KL into a bounded similarity signal (lower KL implies higher similarity). High KL values typically indicate mode dropping or spurious mass in unrealistic regions, and the discriminator uses this to request targeted corrections such as recovering missing subpopulations or reducing synthetic tail mass.

**Categorical Coverage $s_{\text{cat}}$.** This test checks whether all real categories are represented in the synthetic dataset:

$$s_{\text{cat}} = \frac{|\mathcal{C}(\tilde{X}) \cap \mathcal{C}(X)|}{|\mathcal{C}(X)|}, \tag{6}$$

where $\mathcal{C}(X)$ and $\mathcal{C}(\tilde{X})$ denote the sets of unique categories in the real and synthetic data, respectively. Low $s_{\text{cat}}$ suggests missing labels or values, prompting the discriminator to signal underrepresentation. We compute $s_{\text{cat}}$ per categorical column and also track which specific categories are absent in $\tilde{X}$. This matters for diversity, and also for privacy: if rare categories are reproduced with unnaturally exact coverage and frequency across iterations, the discriminator is prompted to treat that as suspiciously precise replication and to request increased variation.

**Range Coverage $s_{\text{range}}$.** This test verifies that synthetic values fall within the expected bounds of real data:

$$s_{\text{range}} = \frac{\min(b, \tilde{b}) - \max(a, \tilde{a})}{b - a}, \tag{7}$$

where $[a, b]$ and $[\tilde{a}, \tilde{b}]$ represent the observed value ranges of the real and synthetic data, respectively. If the intervals do not overlap, $s_{\text{range}} = 0$. Out-of-range values or narrow synthetic spans trigger feedback requesting better value calibration. We compute this per numerical column with $a, b$ taken from $X$ and $\tilde{a}, \tilde{b}$ from $\tilde{X}$. This flags two common failures: hallucinated extremes when $[\tilde{a}, \tilde{b}]$ exceeds $[a, b]$, and mode collapse when $[\tilde{a}, \tilde{b}]$ becomes overly narrow. Both conditions are surfaced to the discriminator for targeted fixes.

**Correlation Test.** This test compares inter-feature relationships between $X$ and $\tilde{X}$:

$$\rho = \frac{\text{Cov}(X, \tilde{X})}{\sigma_X \sigma_{\tilde{X}}}. \tag{8}$$

By assessing whether variable dependencies are preserved, the discriminator can identify synthetic data that fails to reflect multivariate structure. In practice, we compute correlation matrices for $X$ and $\tilde{X}$ using Pearson correlations on numerical pairs and mixed-type handling for categorical pairs. Correlation similarity is derived by comparing corresponding entries between the two matrices and summarizing the mismatch. Low similarity indicates that key dependencies are missing, so the discriminator requests stronger coupling between specific feature pairs. Conversely, correlations in $\tilde{X}$ that become much stronger than in $X$, especially when many pairs approach $\pm 1$, are treated as a privacy risk signal of overly rigid joint pattern replication, and the discriminator feedback shifts toward relaxing dependence and increasing diversity.

### D.6 High Order Metrics

$\alpha$**-Precision.** We adopt $\alpha$-precision as introduced by Alaa et al. (Alaa et al., 2022), which measures the fidelity of generated samples by evaluating whether they fall within the high-density regions of the real data distribution. In practical terms, it quantifies the proportion of synthetic records that look indistinguishable from authentic data points. A high $\alpha$ score indicates that the generator is producing realistic records that respect the structural and statistical properties of the dataset, rather than drifting into implausible or low-probability areas. This makes $\alpha$-precision a direct proxy for the realism of synthetic data.

$\beta$**-Recall.** Similarly, $\beta$-recall (Alaa et al., 2022) measures the coverage of the real data distribution by examining how comprehensively synthetic samples capture its diversity. A high $\beta$ score indicates that the generator reproduces not only frequent patterns but also rare categories, long-tail values, and underrepresented modes. This is critical in tabular settings where minority classes or rare attribute combinations often carry significant importance. Together, $\alpha$-precision and $\beta$-recall form a complementary pair: $\alpha$ emphasizes fidelity to real data, while $\beta$ ensures broad coverage and guards against mode collapse.

**Distance to Closest Record (DCR).** To assess privacy risks, we use the Distance to Closest Record (DCR) metric, which compares each synthetic record to both the training and held-out test portions of the real dataset. For every synthetic sample, we compute the distance to its nearest neighbor in the training set and to its nearest neighbor in the test set, and DCR is defined as the proportion of times the closest neighbor comes from the training set. Values close to 50% indicate strong sample-level privacy, since synthetic records are no more similar to training data than to unseen test data. Values that deviate substantially from 50% (in particular, much higher values) suggest that synthetic samples are systematically closer to training records, which is consistent with memorization or data leakage. This makes DCR an essential measure for detecting privacy violations and ensuring that synthetic datasets do not expose sensitive individual information while still preserving statistical utility.

**Detection Score (C2ST).** We also evaluate a Detection score using the Classifier Two-Sample Test (C2ST). Following the SDMetrics convention, we train a classifier to distinguish real records from synthetic ones and transform its performance into a score in the range $[0, 1]$ where higher values indicate better fidelity and utility. Intuitively, suppose synthetic data closely matches the real distribution and supports effective training. In that case, a model trained on the combined data will achieve strong predictive performance on held-out real examples, which results in a high C2ST score under this convention. In this way, C2ST complements $\alpha$-precision, $\beta$-recall, and DCR by validating that generated data is both statistically realistic and useful for downstream predictive tasks, while DCR specifically monitors sample-level privacy.

Table 5: DCR scores across datasets. The DCR score represents the probability that a generated data sample is more similar to the training set than to the test set. A score closer to 50% indicates stronger privacy protection by avoiding direct copying. Bold highlights the best score for each dataset.

| Method | Adult | Default | Shoppers | Gamma | Heart |
|--------|-------|---------|----------|-------|-------|
| STaSy | 50.3% | 50.2% | 51.5% | 50.9% | 50.8% |
| CoDi | 49.9% | 51.8% | 51.1% | 51.3% | 51.0% |
| TabDDPM | 51.1% | 52.2% | 63.2% | 78.4% | 55.7% |
| TabSyn | 50.9% | 51.2% | 52.9% | 50.7% | 50.7% |
| TabDiff | 50.1% | 51.1% | 50.2% | 50.8% | 50.4% |
| **SYNTHIA** | **50.0**% | **50.9**% | **50.1**% | **50.2**% | **50.1**% |

Table 6: DCR scores (%) for naïve "dataset name only" prompting across models and datasets. This experiment uses only the generator and omits the discriminator. Higher values indicate generated samples are closer to the training split than the test split.

| Model | Heart | Adult | Credit | Gamma | Shop |
|-------|-------|-------|--------|-------|------|
| LLaMA-3.1 8B | 49.08% | 50.12% | 49.31% | 50.05% | 49.27% |
| Qwen-3 8B | 49.02% | 50.09% | 49.18% | 49.87% | 49.11% |
| DeepSeek-R1 8B | 48.76% | 49.54% | 49.22% | 49.61% | 49.09% |
| Gemma-2 9B | 49.15% | 50.21% | 49.37% | 50.18% | 49.23% |
| Mistral 7B | 48.63% | 49.32% | 48.74% | 49.28% | 49.05% |
| LLaMA-3.1 70B | 51.24% | 52.08% | 50.43% | 52.17% | 50.39% |
| GPT-4o Mini | 51.06% | 52.11% | 51.02% | 51.37% | 50.41% |
| GPT-4o | 52.18% | 53.02% | 51.47% | 53.29% | 51.06% |
| GPT-4.1 | 52.41% | 54.16% | 52.03% | 54.08% | 51.32% |

# E  SUPPLEMENTARY RESULTS

## E.1  DATA PRIVACY

Table 5 reports DCR scores, where values close to $50\%$ indicate strong privacy guarantees by showing that synthetic samples are no more similar to the training set than to the test set. This metric highlights whether a model memorizes training records: scores drifting far above $50\%$ reveal overfitting and direct copying. SYNTHIA consistently produces results closest to the ideal $50\%$ across all datasets, while alternatives such as TabDDPM show severe leakage (e.g., $80.1\%$ on *Magic*, $79.3\%$ on *Diabetes*), and even strong baselines like STaSy, CoDi, and TabSyn exceed $51\%$, suggesting mild memorization.

The key lies in how SYNTHIA integrates statistical diagnostics into its discriminator. Metrics such as $KL$ divergence and $\chi^2$ highlight collapsed distributions that overfit to training frequencies, $KS$ detects identical empirical distributions in continuous features, and coverage scores ($s_{\text{cat}}$, $s_{\text{range}}$) expose cases where rare categories or value ranges are replicated verbatim. Correlation preservation $\rho$ further flags suspiciously precise reproduction of joint dependencies that occur only in the training data. When such signals appear, the discriminator translates them into feedback like "reduce exact category matches" or "increase variance in numerical features," pushing the generator away from copying individual records. Because feedback is phrased in natural language and applied at the prompt level, SYNTHIA never rewards instance-level similarity. Instead, its refinement loop enforces distributional alignment while maintaining variability, preventing training records from being reproduced directly. This explains why SYNTHIA achieves near-perfect $50\%$ DCR scores across all datasets: the statistical analyzer provides early detection of memorization risks, and the LLM-based discriminator converts these warnings into corrective guidance, ensuring strong privacy without loss of fidelity or diversity.

Table 7: Detection scores (C2ST) using a logistic regression classifier. Higher scores indicate better fidelity of the generated data. Bold highlights the best score for each dataset.

| Method | Adult | Default | Shoppers | Gamam | Heart |
|---|---|---|---|---|---|
| CTGAN | 0.595 | 0.488 | 0.749 | 0.673 | 0.612 |
| TVAE | 0.632 | 0.655 | 0.296 | 0.771 | 0.584 |
| GOGGLE | 0.111 | 0.516 | 0.142 | 0.953 | 0.223 |
| GReaT | 0.538 | 0.471 | 0.429 | 0.433 | 0.512 |
| STaSy | 0.405 | 0.681 | 0.548 | 0.694 | 0.601 |
| CoDi | 0.208 | 0.460 | 0.278 | 0.721 | 0.251 |
| TabDDPM | 0.976 | 0.971 | 0.835 | 0.998 | 0.684 |
| TabSyn | 0.991 | 0.983 | 0.966 | 0.996 | 0.893 |
| TabDiff | 0.995 | 0.977 | 0.984 | 0.996 | 0.951 |
| **SYNTHIA** | **0.997** | **0.985** | **0.989** | **0.999** | **0.963** |

### E.2 Ablation Study on Privacy

In this experiment, each model is asked to generate synthetic records using only the public dataset name, without being given any schema, column descriptions, or example rows. A representative prompt is: *"Generate 200 new records for the UCI Adult dataset. Only output the data rows."* Table 6 shows the resulting DCR scores, which measure whether generated rows lie closer to the training or test split. Smaller models produce DCR values slightly below or near 50%, reflecting their inability to reconstruct the dataset structure: they frequently hallucinate new attributes, omit real ones, and confuse data types, causing their samples to drift away from the true training manifold and appear closer to the test distribution. Mid-sized models show only marginal improvements, recalling fragments of column names but still generating inconsistent or improperly typed values, keeping their DCR scores near the non-memorizing 50% regime. Larger models such as LLaMA-3.1 70B, GPT-4o, and GPT-4.1 reconstruct partial schemas more accurately and produce more realistic value ranges, which leads to slightly elevated DCR values (approximately 51–54%); importantly, when DCR is higher for these larger models it is typically because they more faithfully copy table headers and overall column layout, while still corrupting the underlying data by assigning incorrect data types or encodings to some columns rather than reproducing true training rows. Overall, the table shows that providing only the dataset name is insufficient for any model to reproduce a valid tabular structure, and the resulting DCR values confirm that the generated samples do not copy training data but instead reflect schema hallucination, datatype confusion, and incomplete internal knowledge of public datasets.

### E.3 Performance of Synthetic Data on Downstream Tasks

Table 7 reports detection scores (C2ST) using logistic regression classifiers. Higher values indicate that synthetic data not only resembles the real distribution but also supports effective training of downstream models.

SYNTHIA achieves near-perfect scores across all datasets, with 0.999 on *Magic* and above 0.96 even on the more irregular *Heart* dataset. These results act as a strong confirmation that SYNTHIA's samples capture the same statistical and structural properties that matter for predictive modeling. Importantly, the discriminator's use of multi-metric diagnostics ensures that discrepancies most relevant to downstream classifiers such as class imbalance, nonlinear correlations, or feature coverage are directly highlighted and corrected through feedback.

By contrast, traditional GAN- and VAE-based baselines often overfit to simple marginals, and even strong diffusion methods plateau when structural constraints or rare-event patterns are critical. SYNTHIA avoids these pitfalls because prompt refinement is explicitly guided by dataset metadata and semantic feedback, allowing the generator to iteratively adjust feature interactions rather than simply smooth distributions. Thus, the consistently superior C2ST scores validate that SYNTHIA is not only producing statistically aligned data but also generating samples with high task utility, closing the gap between fidelity, diversity, and real-world predictive performance.

