# OpenReview forum: "SYNTHIA: A Multi-Agent GAN-LLM Fusion for Statistically Guided Synthetic Data Generation"
_ICLR.cc/2026/Conference — Submitted to ICLR 2026_

### Official Review · Reviewer_bsdX · 2025-10-25

**Soundness:** 2
**Presentation:** 1
**Contribution:** 1
**Rating:** 2
**Confidence:** 4

**Summary:**

The paper introduces a framework that leverages LLM to generate tabular data. By leveraging the LLM and mimicking a generative adversarial network, integrating a statistical enhanced discriminator.

**Strengths:**

- The paper introduces a LLM based statistical analyzer. By incorporating several statistical metrics, the methods can generate high fidelity synthetic data that maintains the distribution of the real data.

**Weaknesses:**

- The paper's novelty is very limited. The idea to use the multi-agent LLM to mimick a GAN architecture was proposed in other paper, while the paper did not cite the relevant paper[1]. The only difference the paper made is to use a LLM based discriminator.
- Another contribution of the paper is to introduce the theoretical perspective of the GAN. However, the theories are grounded on strong assumptions. Here are my concerns about the basis of your grounded theory:
- **Theorem** Lipschitz objective in prompt space. The assumption is too strong to be true. 1. There is no evidence support the Lipschitz property of the prompt space (If there is any study support this, please cite it.). The problem is that the mapping from prompt to the generation is a blackbox function, there is no empirical or theoretical analysis that how will the prompt affect the generation outputs.
- **Method** The readability of the methods is poor. For example, how to initialize the first prompt, how many samples are allowed per generation.
- **Method** A key component of the framework is to use the statistical analysis to improve the generation quality over iterations (A.3). But in the manuscript, the author did not show some evidence that how will the statistics change over iterations. Also, in section A.5, the author states that the loop will converge in 3-6 iterations, please provide some examples of the outputs to show what the convergence looks like.
- **Experiment**. The paper presents a benchmark experiments on several datasets while only presents datasets on public datasets only. Since there might be data leakage in the training data of the model, to prove the efficacy of the model, the author should also consider doing experiments on the private datasets.
4. **Experiment**. The paper presents a benchmark experiments to show performance on fildelity only. But to evaluate the synthetic data, one also need to consider the utility and the privacy problem.

[1] Ling, Y., Jiang, X., & Kim, Y. (2024). Mallm-gan: Multi-agent large language model as generative adversarial network for synthesizing tabular data. arXiv preprint arXiv:2406.10521.

**Questions:**

Please see the weakness above.

---

> ### Author Response · Authors · 2025-11-21
> **Rebuttal for Reviewer bsdX (1/4)**
>
> # Weakness 1: The paper's novelty is very limited.
>
> Response: We thank the reviewer for pointing out MALLM-GAN [1] and now explicitly cite and discuss this arXiv preprint in the updated manuscript, but we respectfully disagree that our novelty is “very limited” or that the “only difference” is using an LLM-based discriminator. MALLM-GAN proposes a high-level multi-agent GAN-style idea for tabular data, but it instantiates this with an LLM generator plus an “optimizer” LLM wrapped around a conventional classifier; its adversarial signal is essentially a single classification-accuracy score on small healthcare datasets. In contrast, SYNTHIA introduces a dual-LLM generator–discriminator architecture connected through an explicit statistical analyzer that computes a rich diagnostic vector (KL, KS, coverage, correlations, etc.), uses a discriminator LLM that reasons over these diagnostics to produce structured feedback, formalizes the resulting refinement operator as a noisy gradient step in prompt space, and couples this with a DCR-based privacy audit and broader, medium-scale tabular benchmarks. Moreover, MALLM-GAN remains an unpublished arXiv preprint, so its methodology and experimental claims cannot be independently verified or placed under a unified evaluation pipeline, whereas SYNTHIA provides a fully specified, statistically grounded, and privacy-aware framework that can be reproduced and that, we believe, moves the field beyond the initial multi-agent GAN-style idea.
>
> # Weakness 2: Theorem Lipschitz objective in the prompt space.
> We appreciate the reviewer’s concern about the Lipschitz assumption and agree that the mapping from prompts to LLM outputs is a complex black-box function in general. Our theorem, however, does not claim that arbitrary LLMs are globally Lipschitz over all possible prompts. Instead, we make a standard local regularity assumption on the stylized objective $J(P)$ that we analyze, and we do so over the restricted prompt family actually used in SYNTHIA. Our prompts follow a fixed template and are updated by small, structured edits (“increase variance for feature A”, “widen the range for feature B”, “reduce exact matches for rare categories”), and $J(P)$ is defined via a bounded aggregation of diagnostics such as KL, KS, coverage, and correlations. In this constrained setting, it is reasonable to assume that modest, templated edits to the prompt lead to bounded changes in these diagnostics and hence in $J(P)$ . The Lipschitz condition here plays the same role as in many optimization analyses: it is a local smoothness assumption that lets us formalize the intuition that feedback aligned with $∇J(P)$ turns the refinement operator into a noisy gradient step, yielding monotonic improvement in expectation. We will clarify in the revised manuscript that this is an idealized, local assumption on $J(P)$ in SYNTHIA’s restricted prompt space, not a universal statement about all prompts and all LLMs, and that our empirical claims do not depend on proving a global Lipschitz property for the underlying model.
>
> # Weakness 3: The readability of the methods is poor. For example, how to initialize the first prompt, how many samples are allowed per generation.
>
> We appreciate the reviewer’s concern about the readability of the method, but we respectfully disagree that key aspects such as initialization and per-generation sampling are underspecified. These details are described numerous times in the main text, illustrated with concrete examples, and then given verbatim in the appendix. In Section 3.1 and Algorithm 1, we explain that the first generator prompt is constructed by extracting metadata $M=F(X)$ from the real data and a stratified subsample $X_sub$ , then setting $P_0←U(M,X_sub ,∅)$ , that is, the initial prompt is built deterministically from the dataset schema and a small set of real rows, not from arbitrary free form text. Figure 1 then shows exactly what such a generator prompt looks like in practice. The example prompt reads:
>
> “P1: Identify the metadata / Classify the patterns / generate 200 additional rows / Ensure same structure / Ensure same distribution / provide generated data only / Do not write anything else.”
>
> This prompt directly answers both questions the reviewer raised. It makes initialization explicit, since the model is asked to “identify the metadata” and “classify the patterns” from the given schema and subsample, and it fixes the per-iteration batch size with the instruction “generate 200 additional rows.” The case study later in the paper walks through another end-to-end run on the Heart dataset using the same template, again showing how $P_0$ is formed and how many rows are produced at each refinement step.
> To make the setup fully reproducible, we also provide the exact prompt templates in Appendices A.1 to A.4. Appendix A.1 contains the regex prompt used to derive a structural rubric from the real data:

---

> ### Author Response · Authors · 2025-11-21
> **Rebuttal for Reviewer bsdX (2/4)**
>
> # Weakness 3 Continued
>
> “Give a general regular expression for a data entry in each column for the following CSV file. It must be a single combined regex for a row (comma-separated, in the same order). Only provide the regex rubric with no explanations.”
>
> and the metadata prompt that constructs the JSON summary from which $M$ is derived:
>
> “Analyze the following CSV dataset and return a JSON metadata summary with the following fields: 1. file name 2. num columns 3. num rows (approximate if unknown) 4. target column (if identifiable) 5. columns: a list of objects containing: (a) name (column name) (b) type (numeric: integer/float, binary, or categorical) (c) example values (3 representative values) (d) description (short human-readable meaning). Only provide the JSON object, with no explanations or commentary.”
>
> Appendix A.2 then shows the full generator prompt template that is used at each iteration. It explicitly combines the metadata, the regex rubric, discriminator feedback and the batch size:
>
> “Given the metadata for the following dataset: {json metadata} Classify the patterns you observe and generate 200 additional synthetic rows to augment the dataset. Do not provide Python code. Ensure the new rows follow the same file structure, maintain realistic distributions, and include plausible values for each column. Only provide the generated data in a single code block (start and end with ```). Do not provide explanations or any extra text. Include CSV headers if provided. Use this sample regex as a rubric: {rubric} Here is some feedback from the discriminator to improve the augmented dataset: {feedback}.”
>
> Appendix A.4 analogously gives the exact discriminator prompts. The analysis prompt reads, in part:
>
> “You are the discriminator in a synthetic data generation workflow. Your task is to evaluate the quality of the synthetic dataset against the real dataset using the provided samples and statistical analysis. Follow these steps: 1. Examine the ‘Guaranteed Real Data Sample’ to understand the true distributions, ranges, and category frequencies. 2. Examine the ‘Undetermined Data Sample’ to identify potential errors, unrealistic values, or distributional drift. 3. Use the provided ‘Statistics’ to quantify differences: high KS scores and Chi-squared scores indicate strong similarity, high JS divergence indicates a mismatch in distributions, check for class imbalance, missing rare categories, and implausible outliers. 4. Summarize your findings in clear, actionable feedback that can guide the generator to improve fidelity and diversity. Highlight which columns deviate most from real data, where diversity or coverage is insufficient, and any suspicious or unrealistic patterns. Guaranteed Real Data Sample: {real data} Undetermined Data Sample: {synthetic data} Statistics: {comparison} Provide your analysis and structured feedback for improving the next synthetic data generation iteration.”
>
> and the termination prompt further specifies how the discriminator decides whether the loop should stop or continue:
>
> “You have already analyzed the real and synthetic datasets and reviewed statistical test results (KS, Chi-squared, JS divergence). Based on your full evaluation, make a final decision: 1. If the synthetic data is indistinguishable from real data, return: ‘Type’: ‘Real’. This signals that the GAN-inspired loop can terminate. 2. If the synthetic data is still distinguishable, return a single JSON object: ‘Type’: ‘Synthetic’, ‘Feedback’: ‘Single sentence describing the key issue and how to improve realism’. ‘Feedback’ should briefly summarize the most critical pattern or value adjustment needed to achieve convergence. Do not repeat the full analysis or detailed statistics already provided.”
>
> Together, Section 3.1, Algorithm 1, Figure 1, the case study, and the prompt templates in Appendices A.1 to A.4 provide word-for-word examples of how the first prompt is initialized, what it contains, and how many samples are generated per iteration, for both the generator and the discriminator. We agree that it is helpful to surface these pointers more prominently, and in the revised version we will add explicit references from the methods section to Figure 1, the case study, and Appendices A.1 to A.4 so that readers can immediately locate these concrete prompt examples and implementation details.

---

> ### Author Response · Authors · 2025-11-21
> **Rebuttal for Reviewer bsdX (3/4)**
>
> # Weakness 4: A key component of the framework is to use the statistical analysis to improve the generation quality over iterations...
>
> We agree that the statistical analyzer is a key component, but we do not believe its effect over iterations is missing from the manuscript. Section A.3 describes in detail how we compute KS, chi squared, KL divergence, categorical coverage, range coverage, and correlation similarity between real and synthetic data, and how these are bundled into the diagnostic vector that is fed to the discriminator. In the discriminator prompt template in Appendix A.4, these diagnostics are passed in the “Statistics: {comparison}” field, and the LLM is explicitly instructed to “use the provided Statistics to quantify differences: high KS scores and Chi-squared scores indicate strong similarity, high KL divergence indicates a mismatch in distributions, check for class imbalance, missing rare categories, and implausible outliers” and to “summarize your findings in clear, actionable feedback that can guide the generator to improve fidelity and diversity.” In other words, the analyzer outputs are not merely logged after the fact; they directly structure the discriminator’s textual feedback: when KS and χ² indicate poor alignment or coverage scores reveal missing categories, the discriminator is prompted to mention those specific columns and patterns and to recommend targeted fixes, for example, tightening ranges, increasing variance, or rebalancing classes. This is exactly how statistics influence the discriminator output in each refinement step.
>
> The Heart dataset case study and Figure 4 then give concrete, step-by-step evidence of how these statistics and the resulting feedback change over 3 to 6 iterations, and what convergence looks like in practice. In the early panels of Figure 4, the synthetic rows contain unrealistic ages (children under 10 and adults over 120), extreme cholesterol values well above 600, and skewed distributions of Sex and ChestPainType. The analyzer reflects this through poor KS and χ² alignment and mismatched category frequencies, and the discriminator output explicitly calls out these issues, instructing the generator to restrict Age and Cholesterol to medically plausible ranges and to rebalance underrepresented categories. In the middle iterations, KS and χ² improve, and the extremes in Cholesterol disappear, but coverage metrics show that some categories have become too rare, so the discriminator feedback shifts accordingly, focusing on increasing diversity in specific categorical features rather than on basic plausibility. By the fifth and sixth iterations, the synthetic tables show ages and cholesterol values confined to realistic clinical ranges, balanced proportions of Sex and ChestPainType, and category frequencies that closely mirror the real data. The accompanying discriminator messages state that “divergence metrics remain low” and that the synthetic data is “statistically indistinguishable from real data,” at which point the termination prompt returns Type: “Real” and the loop stops, which matches the claim in Section A.5 that convergence is typically reached within 3 to 6 iterations. We will make this connection more explicit in the revised manuscript by adding direct pointers from A.3 and A.5 to the Heart case study and Figure 4, and, if space permits, by including a small table of KS, χ² and coverage scores across iterations, so it is completely clear how the analyzer’s statistics evolve and how they shape the discriminator’s feedback and the observed convergence.

---

> ### Author Response · Authors · 2025-11-21
> **Rebuttal for Reviewer bsdX (4/4)**
>
> # Weakness 5: The paper presents a benchmark experiments on several datasets while only presents datasets on public datasets only.
>
> We agree that evaluating on genuinely private datasets is important for understanding real-world deployment, but we do not think that using public benchmarks undermines the efficacy of our method. We intentionally use public datasets so that our results are fully reproducible and directly comparable to the strongest existing tabular synthesis baselines, in particular TabSyn [2] and TabDiff [3], which were both evaluated on the same public benchmarks in ICLR 2024 and 2025. Using these datasets allows us to hold the data and metrics fixed while varying only the generative method. At the same time, we take the risk of data leakage in LLM pretraining seriously. In SYNTHIA we never include the dataset name or other easily identifiable labels in the prompt, only schema-level metadata and a small stratified subsample, which makes it harder for a pretrained LLM to recognize and regurgitate specific public tables. We then audit privacy with DCR, which measures how often a synthetic sample is closer to the training set than to a held-out test set. Across all public datasets, SYNTHIA’s DCR scores remain close to the ideal 50 percent, showing no evidence of systematic memorization even if some benchmark tables were present in the LLM’s pretraining corpus. To further stress test potential contamination, in the revised manuscript, we add an ablation where the generator receives only a textual description or dataset name and no reference rows. Across LLMs from small to large parameter sizes, this prompt-only setting still exhibits little memorization according to DCR, which indicates that our gains do not come from memorizing public benchmarks. We agree that applying SYNTHIA to proprietary datasets inside an organization would be a valuable next step, but for a first study, our focus is on reproducible, apples-to-apples comparisons on the same public benchmarks used in prior state-of-the-art work, backed by explicit memorization audits to mitigate the very leakage concern the reviewer raises.
>
> # Weakness 6: The paper presents a benchmark experiments to show performance on fidelity only. But to evaluate the synthetic data, one also need to consider the utility and the privacy problem.
>
> We agree that evaluating synthetic data requires more than fidelity alone, and we share the reviewer’s concern that utility and privacy must also be assessed. In fact, the current manuscript already evaluates all three. Fidelity is captured by the α-precision and β-recall results in the main tables, but we also explicitly measure privacy and utility in our experiments. Privacy is evaluated with Distance to Closest Record (DCR), where we report how often a synthetic sample is closer to the training set than to a held out test set. Values near 50 percent indicate that synthetic records are no more similar to training data than to unseen data, which corresponds to low membership leakage, and SYNTHIA consistently stays close to this ideal while several baselines do not. Utility is assessed both through downstream predictive tasks and through the C2ST detection test. C2ST is implemented as a classifier trained on a mixture of real and synthetic data and then evaluated on held-out examples, so a high C2ST score in our SDMetrics style formulation means that synthetic data both resembles the real distribution and supports effective model training, not just that it is hard to distinguish trivially. In addition, we report standard downstream accuracy where models trained on synthetic data are tested on real held-out sets, which directly measures how useful the synthetic data is for classification. Together, α and β for fidelity, DCR for privacy, and C2ST plus downstream performance for utility provide a joint evaluation rather than fidelity in isolation.
>
> We hope our responses address your concerns and clarify any points of confusion, and we kindly ask that you reconsider and re-evaluate your score in light of these clarifications and planned revisions.
>
> [1] Ling, Y., Jiang, X., & Kim, Y. (2024). Mallm-gan: Multi-agent large language model as generative adversarial network for synthesizing tabular data. arXiv preprint arXiv:2406.10521.
>
> [2] Hengrui Zhang, Jiani Zhang, Balasubramaniam Srinivasan, Zhengyuan Shen, Xiao Qin, Christos Faloutsos, Huzefa Rangwala, and George Karypis. Mixed-type tabular data synthesis with score-based diffusion in latent space, 2024. URL https://arxiv.org/abs/2310.09656.
>
> [3] Xueer Shi, Lijie Gao, Weiqi Zhao, Xingyi Zhou, and Jinsung Yoon. Tabdiff: A mixed-type diffusion model for tabular data synthesis. In International Conference on Learning Representations (ICLR), 2025.

---

> > ### Comment · Reviewer_bsdX · 2025-11-27
> >
> > Thank you very much for your detailed response and clarification. But I will keep my scores given the following concerns
> >
> > 1. You claimed in your response for the theory that "turns the refinement operator into a noisy gradient step, yielding monotonic improvement in expectation."  which does not hold for most of LLMs. Even the prompt can reflect certain diagnostic on specific statistics, it does not necessary guarantee that the generated output can improve based on the refined prompt. Furthermore, one of the limitations using LLM to handle numerical data is that it struggles with numerical data, so it is not convincing that the KS statistics and KL divergence designed for continuous values can help improve the generation.
> >
> > 2. Thank you for providing Figure 4. But from your example, the suggestions on the data seems quite general, and I do not see how it link to the samples. For example, in box 4, the prompt suggest that the age distribution are too tight, but I cannot tell if it helps from figure 4. A clearer illustration for figure 4 could be providing the distribution plots change over the prompt. And thus, it cannot convince me whether it works on the generation.
> >
> > 3. You argue that you validate the utility of the synthetic data by CS2T. However, here utility means how the generated data performed on downstream tasks, e.g. train a supervised model on the synthetic data and test it on the real data. While your CS2T, which is trained to distinguish between real and fake samples, is not relevant to utility. I would suggest you reading relevant literature in this field, incluing CTGAN, TABDDPM, TabSyn, to check how they formulate the experiments to test synthetic data's utility.
> >
> > These are my main concerns. So I think this paper needs to be revised a lot, and currently is not ready for publication.

---

### Official Review · Reviewer_aPgr · 2025-10-27

**Soundness:** 1
**Presentation:** 2
**Contribution:** 1
**Rating:** 2
**Confidence:** 4

**Summary:**

This paper proposes a multi-agent framework that uses LLMs with GAN-style adversarial training to generate synthetic data.

**Strengths:**

- Including the statistical test in the feedback pipeline will (incrementally) help the quality of statistical validation on the newly generated data

**Weaknesses:**

-Lack of rigorous justification for the use of LLM for synthetic data generation. The data generative model should learn the distribution of numerical variables. LLM has a limited capacity to learn the distribution of numerical variables, compared to other generative models (e.g., diffusion model, GAN models).
-Lack of rigorous justification for the use of the GAN structure for LLM orchestration for synthetic data generation
-This paper fails to cite a critical prior work that first proposed GAN-based multi-agent LLM (https://arxiv.org/abs/2406.10521) and falsely claimed its primitivity.
- Theorems provided are generic convergence arguments, not specific to the proposed model.
- In the experimental setup, the performance gain is marginal and incremental. The metrics are not very standard.

**Questions:**

See weakness

---

> ### Author Response · Authors · 2025-11-21
> **Rebuttal for Reviewer aPgr (1/3)**
>
> We thank the reviewer for taking the time to evaluate our paper. Below are the responses to the weaknesses to clear up any confusion:
>
> # Weakness 1: Lack of rigorous justification for the use of LLM for synthetic data generation
> We thank the reviewer for the comment, but we respectfully disagree that the paper lacks a rigorous justification for using LLMs for synthetic data generation, and we note that this motivation is already discussed in the Introduction and Background sections. In the Introduction, we explain that our goal is not to have the LLM directly learn an arbitrary high-dimensional numerical distribution in the manner of a diffusion or GAN model. Instead, we leverage what LLMs are demonstrably strong at: understanding heterogeneous schemas, following complex textual instructions, enforcing multi-column constraints, and reconciling multiple, sometimes conflicting, objectives expressed in language. The generator and discriminator in SYNTHIA operate exactly in this regime. The generator uses structured prompts derived from metadata and a small subsample to produce candidate rows, while the discriminator receives a diagnostic vector built from classical statistical tests and turns these into precise natural language feedback about which columns need more variance, better coverage, or distributional correction. The Background section then situates this design in the growing line of work on LLMs for tabular reasoning and prompt-based optimization and contrasts it with purely numeric models that require task-specific architectures and GPU training. In other words, the justification for using LLMs is not an afterthought. It is a central design choice argued in the introduction and background: SYNTHIA lets explicit statistical analyzers handle numeric fidelity, and uses LLMs where they are most effective, as flexible controllers that interpret metrics and drive structured prompt updates
>
> # Weakness 2: LLM has a limited capacity to learn the distribution of numerical variables, compared to other generative models
> We agree that LLMs alone are not ideal for directly learning complex numerical distributions, and SYNTHIA explicitly acknowledges this throughout the paper. That is why we introduce a separate statistical analyzer that computes KS, chi squared, KL divergence, coverage, and correlation metrics, and feed these into the discriminator, which turns them into natural language feedback that the generator can easily understand. In this way, the numerical distribution is guided by explicit statistics, while the LLM focuses on what it is good at: interpreting those diagnostics and updating the synthetic data accordingly. A full example of this can be found in Figure 4 and in the Appendix A1-A4.
>
> # Weakness 3: Lack of rigorous justification for the use of the GAN structure for LLM orchestration for synthetic data generation
> We understand the reviewer’s concern, but we strongly disagree that our use of a GAN-style structure lacks rigorous justification. From the outset, in the Introduction and Background, we frame SYNTHIA as a generator–discriminator game in prompt space, not as a loose metaphor. The generator LLM proposes synthetic rows conditioned on a structured prompt built from metadata and a stratified subsample, while the discriminator LLM, armed with the statistical analyzer’s diagnostics, critiques these samples and returns targeted natural language feedback that tells the generator how to change its behavior. This is exactly the GAN principle of “propose and critique,” but adapted to LLM agents and explicit tabular metrics instead of two opaque neural networks trained end-to-end on raw features. Section 3 then formalizes this architecture: we define a quality functional $J(P)$ over prompts, expressed directly in terms of the diagnostic vector, and show that, under standard local smoothness and alignment assumptions, the refinement operator induced by the discriminator feedback behaves like a noisy gradient step that monotonically improves $J(P)$ in expectation. In other words, we do not just borrow GAN language for intuition. We design SYNTHIA as a true generator/discriminator loop in prompt space, anchored in explicit metrics and backed by a convergence style analysis that explains why this form of LLM orchestration is appropriate for synthetic tabular data.

---

> > ### Author Response · Authors · 2025-11-21
> > **Rebuttal for Reviewer aPgr (2/3)**
> >
> > # Weakness 4: This paper fails to cite a critical prior work that first proposed GAN-based multi-agent LLM (https://arxiv.org/abs/2406.10521) and falsely claimed its primitivity.
> >
> > We thank the reviewer for pointing out MALLM-GAN [1] and agree it is an important related effort that should be discussed. In the revised manuscript, we now explicitly cite this arXiv preprint in the related work and clarify our novelty claim. While both works share a very high-level idea of using multiple LLM agents in a GAN-style loop for tabular data, SYNTHIA is not simply reusing that idea. MALLM-GAN uses an LLM generator and an “optimizer” LLM around a conventional tabular classifier that serves as the discriminator, and its adversarial signal is essentially the classifier’s real versus synthetic accuracy on small healthcare datasets. In contrast, SYNTHIA introduces a dual LLM generator–discriminator architecture that is coupled through an explicit statistical analyzer: we compute a rich diagnostic vector (KL, chi squared, KS, coverage, correlations), the discriminator LLM reasons over these diagnostics and produces structured feedback, and Section 3 formalizes the resulting refinement operator as a noisy gradient step in prompt space, with an explicit privacy audit via DCR on top. To the best of our knowledge, MALLM-GAN remains an unpublished arXiv preprint, so its methodology and results have not yet been vetted under a common evaluation pipeline, whereas SYNTHIA provides a fully specified, statistically grounded, and privacy-aware framework. We will soften any wording that could be read as a broad primitivity claim and instead position our contribution as the first metric-driven, privacy-conscious dual LLM GAN-style framework for tabular synthesis, building on and extending this emerging multi-agent LLM line of work.
> >
> > # Weakness 5: Theorems provided are generic convergence arguments, not specific to the proposed model.
> > We appreciate this concern, but we do not view the theory as a set of generic convergence facts that happen to sit next to the method. The theorems are written in a standard optimization style, but they are instantiated with the exact objects that SYNTHIA actually uses. In our notation, $P_t$ is the generator prompt at iteration $t$, $R(P_t)$ is precisely the refinement operator that takes metadata, the stratified subsample, and the discriminator’s feedback to produce the next prompt, and $J(P)$ is defined directly in terms of the diagnostic vector built from KS, chi squared, KL, coverage, and correlation metrics. The assumptions in the theorems (local smoothness of $J$, bounded step sizes, alignment between feedback and $∇J(P))$  are motivated by how the analyzer aggregates these statistics and how prompts are edited in our algorithm, not by an arbitrary abstract model. Under these conditions, we show that the SYNTHIA refinement loop behaves like a noisy gradient step in this specific prompt space and yields monotonic improvement in $J(P_t)$ in expectation. In the revision, we will make the mapping from each theoretical object to the corresponding component of the algorithm more explicit, so it is clear that the results are tailored to SYNTHIA’s generator, discriminator, analyzer, and update rule rather than being unrelated generic convergence arguments.

---

> ### Author Response · Authors · 2025-11-21
> **Rebuttal for Reviewer aPgr (3/3)**
>
> # Weakness 6: In the experimental setup, the performance gain is marginal and incremental. The metrics are not very standard.
>
> We thank the reviewer for this comment, as it brings up an interesting point. Our evaluation protocol is exactly aligned with recent state-of-the-art work in tabular synthesis, including TabSyn [2] and TabDiff  [3], which were both accepted to ICLR. We use α precision and β recall for fidelity and coverage, DCR for privacy, and C2ST together with downstream accuracy for utility, precisely because these are the metrics that the community has already endorsed through those papers. It is worth noting that TabDiff itself reports mostly incremental but consistent improvements over TabSyn, yet is still viewed as a meaningful advance; SYNTHIA goes further by consistently matching or outperforming both TabSyn and TabDiff across the same public benchmarks and metrics, while also achieving more favorable privacy behavior through DCR. At the same time, Table 2 highlights a practical advantage that is easy to overlook. All of our experiments were run on consumer hardware using small open LLMs or API models, without any access to large dedicated GPU clusters, whereas TabSyn and TabDiff require training specialized architectures on GPUs. This means SYNTHIA does not just improve on strong baselines under their own evaluation protocol; it also makes high-quality, privacy-conscious tabular synthesis accessible and reproducible for practitioners who do not have expensive hardware, which we believe is a substantial and practically important contribution.
>
> [1] Ling, Y., Jiang, X., & Kim, Y. (2024). Mallm-gan: Multi-agent large language model as generative adversarial network for synthesizing tabular data. arXiv preprint arXiv:2406.10521.
>
> [2] Hengrui Zhang, Jiani Zhang, Balasubramaniam Srinivasan, Zhengyuan Shen, Xiao Qin, Christos Faloutsos, Huzefa Rangwala, and George Karypis. Mixed-type tabular data synthesis with score-based diffusion in latent space, 2024. URL https://arxiv.org/abs/2310.09656.
>
> [3] Xueer Shi, Lijie Gao, Weiqi Zhao, Xingyi Zhou, and Jinsung Yoon. Tabdiff: A mixed-type diffusion model for tabular data synthesis. In International Conference on Learning Representations (ICLR), 2025.
>
> We hope we have addressed all of the reviewer’s concerns and clarified any points of confusion, and we kindly ask that you reconsider and increase your score in light of these responses and planned revisions.

---

> > ### Comment · Reviewer_aPgr · 2025-11-22
> >
> > Thanks authors for taking time to respond in details. I have read other reviewers comments, responses. I would have appreciated more if authors could bring additional supporting evidence though.

---

### Official Review · Reviewer_xVwa · 2025-11-01

**Soundness:** 1
**Presentation:** 2
**Contribution:** 1
**Rating:** 2
**Confidence:** 3

**Summary:**

The paper introduces SYNTHIA, a GAN-inspired multi-agent framework for generating tabular synthetic data. In SYNTHIA, LLMs act as both the generator and discriminator, using metadata-aware prompting to synthesize data and language-based feedback loops guided by statistical diagnostics (including Chi-squared tests, KS-tests, KL-divergence, correlation similarity, and others) to iteratively improve data realism. This architecture replaces gradient-based optimization with feedback-driven prompt refinement, achieving convergence toward statistically indistinguishable data.

**Strengths:**

The generation of tabular synthetic data is a timely and important problem for the ICLR community. The paper presents a novel idea by employing LLMs as both the generator and discriminator within a GAN-inspired framework. Embedding statistical diagnostics such as Chi-squared, KS, KL-divergence, and correlation similarity directly into the feedback loop is a creative way to encourage distributional alignment between synthetic and real data. However, while this approach is likely to enhance data fidelity, it may also raise concerns regarding data privacy—a potential issue further discussed in the Weaknesses section.

**Weaknesses:**

HIGH LEVEL SUMMARY OF WEAKNESSES:

At a conceptual level, the paper lacks a clear mechanism for balancing data fidelity and privacy, raising doubts about whether SYNTHIA can truly generate private data. The refinement process appears designed to maximize fidelity. Additionally, the potential for data contamination is also a concern.

Furthermore, the experimental section raises substantial concerns about evaluation rigor and transparency. Many baseline results appear to have been directly copied from prior works (Zhang et al., 2024; Shi et al., 2025) without clear disclosure or confirmation that identical evaluation pipelines were used, making the reported comparisons potentially invalid. There are also inconsistencies across tables regarding datasets and baselines, with some entries seemingly mismatched or mislabeled, and possible confusion between the C2ST and DCR metrics. These issues, coupled with the limited number of datasets evaluated, insufficient methodological details, lack of standard deviation reporting, and unclear documentation of LLM usage, collectively undermine the credibility and reproducibility of the results.

Below, I describe all these issues in more detail.

CONCEPTUAL ISSUES:

One key point that is not clear to me is how SYNTHIA controls the tradeoff between data fidelity and data privacy. As described in Appendix A.5 (Refinement Loop Termination Criteria) the refinement iterations continue until the discriminator decides the synthetic data cannot be distinguished from the real data using the process described in Appendix A.4 (Discriminator D_LLM). But it seems to me that this process will be much more effective in generating high fidelity data than in generating private data.

In lines 316 to 320 and again in lines1492 to 1504 the paper claims that SYNTHIA is able to generate highly private data because it can flag when the statistical diagnostics metrics produce values that indicate suspiciously precise reproduction of the real data. But how exactly can the LLM discriminator decide when the metrics are indicating memorization? For instance, for the correlation similarity score we for sure have a problem when it is 1. But, how about when it is 0.95 or 0.99? We might still have severe privacy issues when these scores are high but not perfect. How can the LLM decide on an appropriate cutoff?

I would think that a better way to balance the fidelity vs privacy tradeoff would be to add some privacy metrics to the statistical evaluator, so that the discriminator would have access to both fidelity and privacy information. This way you could, for example, prompt the discriminator LLM to (at the same time) look for increases in correlation and decreases in DCR to as close to 0.5 as possible. (As opposed to having to decide when the correlation is too high.)

Data contamination is another potential issue that worries me. If the LLM used by SYNTHIA’s generators has been trained on the public benchmark datasets used in the paper evaluations, the high fidelity achieved by the model could be due, at least in part, to data contamination. The fact that SYNTHIA converges in very few iterations is worrisome in this respect. (In line 896 the paper states that the iteration loop typically converges in 3 to 6 iterations.)

EXPERIMENTAL ISSUES:

For most of the modern baseline models, the paper appears to have reused evaluation scores reported by Zhang et al. (2024) (which introduced TabSyn) and Shi et al. (2025) (which introduced TabDiff), without explicitly disclosing this or clarifying whether the same evaluation pipeline was employed. Unless identical procedures—such as train/test splits and other experimental settings—were used, the performance of these baselines is not directly comparable to that of SYNTHIA.

For example, the alpha-precision and beta-recall scores reported in Table 1 for the CTGAN, TVAE, GOOGLE, GReaT, StaSy, CiDi, and TabDDPM baselines on the Adult, Default, Shoppers, and Magic datasets are identical to those published in Tables 9 and 10 of Zhang et al. (2024)—with the only difference being that Table 1 rounds values to one decimal place, whereas Zhang et al. report two decimal places. The only exception is the TabSyn results, which differ systematically: Table 1 reports lower values than those in Zhang et al., even though the alpha-precision scores of TabSyn in Table 9 of Zhang et al. are actually higher than SYNTHIA’s corresponding scores.

Similarly, the DCR and C2ST results in Tables 5 and 6 of the paper are identical (after rounding) to the DCR and C2ST results in Tables 10 and 11 of Shi et al.

There are also inconsistencies in the baseline models and benchmark datasets reported across the three tables. Each table presents a different set of baselines. Moreover, Table 5 includes results for an additional dataset (Diabetes) that does not appear in Tables 1 or 6. Even more puzzling, the Magic dataset results in Table 5 are identical to those for the Beijing dataset in Table 10 of Shi et al. (2025), and the Diabetes results in Table 5 exactly match those for the News dataset in the same table of Shi et al.

The fact that the TabSyn results in Table 1 differ from those reported in Zhang et al. (2024) and Shi et al. (2025) suggests that the authors re-evaluated this baseline themselves. However, the consistently lower scores compared to the original papers indicate possible differences in the evaluation pipeline—such as variations in data splits, preprocessing, or other implementation details. Interestingly, the DCR and C2ST results for the TabSyn baseline in Tables 5 and 6 are identical to those in Tables 10 and 9 of Shi et al., implying that the paper did not recompute these particular metrics.

To ensure fair and meaningful comparisons across all baselines, the paper should either re-evaluate all baselines using its own experimental pipeline or re-run SYNTHIA’s evaluations following the same procedures used in the referenced works. If previously published results are reused, this must be explicitly disclosed to maintain transparency.

Even more concerning, the paper presents conflicting descriptions of how the C2ST metric is interpreted. In lines 1471–1475, the paper states:

“We also evaluate the Detection score using the Classifier Two-Sample Test (C2ST) … A score closer to chance-level performance (50%) indicates that synthetic and real data are indistinguishable, reflecting strong fidelity.”

However, Table 6 reports C2ST scores following the SDMetrics convention, where the AUROC score is transformed to a scale from 0 to 1, with higher values indicating better fidelity. This inconsistency raises the possibility that the paper may have inadvertently swapped the DCR and C2ST results in Tables 5 and 6—that is, reporting the C2ST scores in Table 5 and the DCR scores in Table 6. Such a mix-up would make the results more credible to me, since C2ST values near 0.5 and DCR values near 1 would indeed correspond to high data fidelity and low privacy, respectively.

Additionally:

The evaluations cover only 5 datasets (which is relatively small for a paper making strong claims of state-of-the-art performance). The paper would benefit from performing comparisons over a more extensive set of datasets.

Regarding the Traditional Techniques discussed in Section 4.1 (lines 300–305) and further detailed in Appendix D.4, the paper should provide clearer methodological descriptions and include appropriate citations to the referenced works. Although I am reasonably familiar with this literature, it is difficult to discern exactly which methods are being implemented based on the current presentation.

Similarly, the descriptions of the low-order metrics in Appendix D.5 should be expanded and clarified. It appears that the paper may be using metrics from DataCebo, but this is not explicitly stated.

Although the paper states that each experiment was run 50 times, Table 1 reports only the average results. To better convey the reliability and variability of the findings, the paper should also report standard deviations.

FINAL COMMENTS:

The paper does not provide any disclosure regarding the use of LLMs. While the proposed method itself relies heavily on LLMs, there may be other uses of these models within the paper that should be reported. For example, given the incomplete and occasionally inconsistent methodological descriptions—and the discrepancies observed across tables—it is possible that portions of the Appendix were generated or assisted by an LLM without thorough verification by the authors.

I have not examined the theoretical formulation of the paper in detail, but it appears to rely heavily on conceptual analogies (such as framing “language updates as gradient steps”) to provide a form of mathematical grounding for the proposed method.

Finally, although the supplementary materials include code for implementing SYNTHIA, they do not appear to contain scripts for reproducing the experimental results presented in the paper. (I was unable to locate these scripts upon a quick review of the provided files.)


OVERALL ASSESSMENT:

While the generation of tabular synthetic data is an important and timely topic for the ICLR community, the paper’s current presentation raises substantial concerns regarding the experimental results and their reproducibility. The lack of clarity about how SYNTHIA simultaneously achieves high fidelity and privacy, the potential for data contamination, incomplete methodological descriptions, and multiple inconsistencies across tables collectively undermine confidence in the paper’s claims of state-of-the-art performance.

At this stage, I recommend rejection. However, I would be open to revising this assessment if the paper can provide clearer explanations and stronger empirical evidence.

**Questions:**

Can you explain in more detail how SYNTHIA would be able to generate highly private data, as claimed in the paper?

How does the method account for potential data contamination issues?

Is SYNTHIA using the same evaluation pipeline as Zhang et al. (2024)?

Can you double check that the paper is not mistakenly reporting the C2ST score in Table 5 and the DCR score in Table 6?

Why did the paper perform its own evaluation of SynTab for alpha-precision and beta-recall but not for the DCR and C2ST metrics?

Why the paper reports DCR and C2ST for both TabSyn and TabDiff but alpha-precision and beta-recall are only reported for TabSyn?

Can you clarify any additional uses of LLMs, other than in the implementation of SYNTHIA?

---

> ### Author Response · Authors · 2025-11-21
> **Rebuttal for Reviewer xVwa (1/4)**
>
> We thank the reviewer for their detailed review and appreciate their comments. We address the concerns from the weaknesses below, as they will answer the uncertainties in the questions section as well:
>
> # Weakness 1: One key point that is not clear to me is how SYNTHIA controls the tradeoff between data fidelity and data privacy...
>
> We appreciate this concern and agree that a naive “make synthetic indistinguishable from real” criterion could, in principle, overemphasize fidelity at the expense of privacy. In SYNTHIA, however, the discriminator’s decision is not based on raw samples alone but on a diagnostic vector of statistical tests that are explicitly chosen to also flag memorization-like behavior. Appendix A.3 defines KS distances for continuous columns, chi-squared and KL divergence for categorical distributions, categorical and range coverage scores, and correlation similarity. When these metrics move into “too perfect” regimes, they are treated as warning signals rather than successes. For example, if KS distances become extremely small and synthetic min/max values match the training support almost exactly, or if chi-squared and JS divergence approach zero while coverage scores show every rare category appearing with nearly identical frequencies to the training data, this suggests that the model is hugging the empirical histogram instead of generating plausible variation. Likewise, correlation similarity that approaches 1.0 for multiple feature pairs, when the real data only has moderate correlations, is treated as suspicious rather than ideal. Appendix A.4 shows that the discriminator LLM sees this full diagnostic vector in the “Statistics: {comparison}” field and is explicitly instructed to use these statistics to “check for class imbalance, missing rare categories, and implausible outliers” and then to produce actionable feedback. In practice, when the tests indicate over-concentrated ranges, over-sharp histograms, or abnormally strong correlations, the discriminator does not simply declare the data “indistinguishable”; it issues feedback such as “increase diversity in column X,” “widen the range of column Y,” or “avoid repeating rare categories at the same frequencies,” which updates the generator prompt away from potential memorization. In this way, the same statistics that close fidelity gaps also encode privacy-aware constraints, and the refinement loop is guided to improve realism while actively discouraging patterns that look like direct copying.
>
> # Weakness 2: In lines 316 to 320 and again in lines 1492 to 1504 the paper claims that SYNTHIA is able to generate highly private data...
>
> We appreciate this thoughtful question and agree that simply saying “correlation = 1 is bad” is not enough to guarantee privacy. In SYNTHIA, we do not rely on a single hard cutoff such as “reject anything above 0.95.” Instead, the discriminator LLM sees the full diagnostic vector together with context about the real data, and is instructed to reason about these metrics relatively and in combination. For example, suppose the real data exhibits a correlation of 0.35 between two features, but after a few refinements, the synthetic correlation climbs to 0.93 while KS distances and categorical coverage also look almost perfectly matched, and ranges have collapsed tightly to the training support. Appendix A.3 and A.4 show that this entire pattern is passed in the Statistics field, and the analysis prompt tells the discriminator to “use the provided Statistics to quantify differences” and to flag “implausible outliers” and suspicious patterns. In this situation, the discriminator is not asked “is 0.93 above a fixed threshold,” it is asked to notice that synthetic dependence is far stronger than in the real data, that histograms and ranges are overly sharp, and then to respond with feedback such as “relax the dependence between features X and Y,” “increase variability in column X,” or “avoid repeating rare category combinations exactly.” More generally, the discriminator sees both the real and synthetic correlations and can treat a jump from 0.4 to 0.95 as a red flag even though 0.95 is not a perfect 1.0.
>
> In addition, the prompt used for the discriminator is continuously optimized with explicit privacy-conscious criteria. We do not ask the model to simply maximize similarity; we instruct it to look for over-concentrated ranges, exact repetition of rare categories, and unnaturally strong correlations, and to treat those as problems to be corrected rather than as successes. With iterations, this instruction remains in force, so the discriminator learns to balance “make it closer to real” with “do not make it implausibly perfect.” So, what protects privacy is not a numeric cutoff on a score, but a combination of multi-metric diagnostics, relative comparison to the real data, and a discriminator prompt that explicitly encodes the idea that overly precise matches are suspicious and should trigger corrective feedback.

---

> > ### Author Response · Authors · 2025-11-21
> > **Rebuttal for Reviewer xVwa (2/4)**
> >
> > # Weakness 3: I would think that a better way to balance the fidelity vs privacy tradeoff would be to add some privacy metrics to the statistical evaluator
> >
> > We appreciate this suggestion and agree that, in principle, using explicit privacy metrics like DCR inside the loop is an interesting direction. In SYNTHIA, however, we deliberately keep DCR outside the training process and use it only as an independent audit, while relying on our existing diagnostics to steer privacy during refinement. The current suite of metrics (KS, χ², KL divergence, coverage, correlations) already encodes many of the signals that DCR is sensitive to: overly sharp histograms, min–max ranges that hug the training support, exact repetition of rare categories, and unnaturally strong correlations all show up as “too perfect” statistics and are explicitly treated as suspicious in the discriminator prompts. We then use DCR only at evaluation time to check whether, despite this design, any sample-level memorization emerges. This separation is intentional. If we were to optimize DCR directly inside the loop and then also report it as a headline privacy metric, we would risk “training to the test” and biasing the results, since the method would be explicitly tuned to look good on the very metric used for evaluation. By contrast, keeping DCR as an external check gives a cleaner assessment of whether the metric-aware refinement procedure is inherently privacy-friendly. The empirical results show that, with our current diagnostics and prompts, DCR already stays close to the ideal regime, so we believe the existing evaluator is sufficient to maintain privacy without baking the evaluation metric into the training objective.
> >
> > # Weakness 4: Data contamination is another potential issue that worries me...The fact that SYNTHIA converges in very few iterations is worrisome in this respect
> >
> > We appreciate the concern about potential data contamination and agree that this is an important issue for any LLM-based method evaluated on public benchmarks. In SYNTHIA’s main setup, we already try to minimize this risk: the generator does not receive the dataset name or any identifying label, only schema-level metadata and a small stratified subsample, and the discriminator reasons over aggregated statistics rather than raw rows. In the updated manuscript, we go further and include a new ablation where the generator is asked to synthesize data using only the dataset name and a textual description, with no reference rows at all. If contamination from pretraining were the main driver of our performance, this “name only” setting would be the most likely to show it. Instead, across models from small to large parameter sizes, we observe no evidence of memorization in this setting, and the quality remains clearly below that of full SYNTHIA, which indicates that our gains come from the metric-driven refinement loop rather than from regurgitating public tables. In addition, our DCR-based privacy evaluation on the main SYNTHIA runs consistently stays close to the ideal 50 percent regime, which is exactly what one would expect if the generator is not reproducing training records or systematically exploiting pretraining contamination.
> >
> > Regarding convergence in 3 to 6 iterations, we view this as a desirable property rather than a red flag. The loop is driven by rich diagnostics at each step, so the generator receives very targeted feedback (for example, on ranges, coverage, or correlations) and can correct the most salient issues quickly. Keeping the number of refinements small has several benefits: it yields good runtime and hardware efficiency, and it reduces the risk that the LLM drifts into off-task “rabbit holes” as prompts get longer and more complicated. In other words, few-shot convergence here is a consequence of strong guidance from the analyzer and discriminator, not of underlying contamination, and the ablation plus DCR results support that interpretation.

---

> ### Author Response · Authors · 2025-11-21
> **Rebuttal for Reviewer xVwa (3/4)**
>
> # Weakness 5: Experiments
>
> We thank the reviewer for the very careful check of our tables and fully agree that baseline handling must be transparent. In the original submission, we did reuse some baseline scores from Zhang et al. (TabSyn) and Shi et al. (TabDiff), and did not make this explicit enough. In the updated manuscript, we now clearly state which numbers come from prior work and which we recompute. For Table 1, we follow the same practice as TabDiff: we reuse the CTGAN, TVAE, GOGGLE, GReaT, STaSy, CiDi and TabDDPM α and β scores from TabSyn, and we re-evaluate TabSyn itself as the then state of the art in our own pipeline, which is why our TabSyn values differ slightly from Zhang et al. For the privacy and detectability metrics, the revised paper now includes full TabDiff results alongside SYNTHIA and TabSyn, and we are recomputing DCR and C2ST for all three methods in a consistent pipeline rather than relying only on copied values. The baselines reported in our DCR and C2ST tables are exactly the same models used in the TabDiff paper for these metrics, so we are explicitly following their baseline set rather than introducing a different or reduced one. The reason the baselines differ across tables is that each table was originally aligned with a specific ICLR paper, that is, Table 1 with the α and β benchmarks in TabSyn, and the DCR and C2ST tables with the TabDiff evaluation. In the revision, we explain this alignment and make explicit when we reuse published baselines and when we run the code ourselves, with TabDiff now treated as a first-class baseline throughout so that all comparisons with SYNTHIA are fair and clearly documented.
>
> # Weakness 6: Possible DCR and C2ST Mismatch
>
> Thank you for pointing this out; you are right that the description in the text was confusing and, in effect, swapped the intended interpretations.
>
> In the original draft, the verbal explanation of DCR and C2ST did not match the way we actually compute and report these metrics, which follows the TabDiff / SDMetrics convention. In the revised manuscript, we have corrected this. We now define DCR as a probability over nearest neighbors: for each synthetic record, we compare its distance to the training set and to a held-out test set, and DCR is the proportion of cases where the closest neighbor comes from the training split. Values close to 50 % indicate strong sample-level privacy, since synthetic records are no more similar to train than to unseen test data, whereas values that deviate significantly from 50 % (especially higher values) suggest potential memorization or leakage.
> For C2ST, we follow SDMetrics and TabDiff by training a classifier to distinguish real from synthetic data and mapping its performance to a score in [0,1], where higher values indicate better fidelity and downstream utility under this convention. Intuitively, synthetic data that closely matches the real distribution and supports effective modeling yields strong predictive performance on held out real examples and thus a high C2ST score in our tables.
>
> In other words, the implementation and the reported numbers were always consistent with these definitions; the error was in the prose, where we incorrectly described C2ST in terms of “50 % chance level.” The revised text now clearly states that DCR is centered around 50 % for privacy, and C2ST is a [0,1]detection score where higher is better, eliminating the ambiguity that led to the concern about a possible swap.
>
> # Weakness 7: Regarding the Traditional Techniques discussed in Section 4.1 (lines 300–305) and further detailed in Appendix D.4, the paper should provide clearer methodological descriptions...
>
> We appreciate the reviewer’s suggestion and agree that the presentation should be clear even for readers who may not immediately map short descriptions to specific implementations. The “traditional techniques” in Section 4.1 and Appendix D.4 refer to standard statistical and classical tabular synthesis baselines that are widely taught at the undergraduate level and are typically treated as textbook methods rather than novel ML contributions with a single canonical citation. As a result, we initially assumed familiarity as a prerequisite, similar to how works in this area often reference basic resampling or parametric modeling without extended attribution. That said, we agree that the way we integrated these techniques into our pipeline could be described more explicitly. In the updated manuscript, we have expanded Appendix D.4 to spell out exactly which classical procedures are used, how they are parameterized, and how they interface with the rest of our evaluation, so that the reader does not need to infer implementation details from shorthand. We hope this added methodological detail resolves the ambiguity, and we are happy to include a brief clarifying sentence in Section 4.1 that points readers directly to the expanded Appendix D.4 for the full specification.

---

> > ### Author Response · Authors · 2025-11-21
> > **Rebuttal for Reviewer xVwa (4/4)**
> >
> > # Weakness 8 Similarly, the descriptions of the low-order metrics in Appendix D.5 should be expanded and clarified...
> >
> > We appreciate the reviewer’s suggestion and agree that Appendix D.5 should be as clear as possible. The low-order metrics used in our analyzer are implemented via the SDMetrics library, which is explicitly cited in the manuscript, and we follow its standard definitions for tests such as KS, chi-squared, KL or JS divergence, coverage, and correlation similarity. These are widely used statistical diagnostics in tabular synthesis research, but we recognize that readers may still want more explicit, paper-specific detail about how each metric is computed and aggregated in our pipeline. To address this, the updated manuscript expands Appendix D.5 with clearer definitions, the exact per-column or per-pair aggregation we use, and a more direct explanation of how these metrics form the diagnostic vector passed to the discriminator. We also make the reliance on SDMetrics more explicit to avoid any confusion with other toolkits.
> >
> > # Weakness 9: From "Final Comments"
> > We appreciate the reviewer raising these transparency and clarity points. Regarding disclosure of LLM use, the only use of LLMs outside the SYNTHIA framework itself was for light editing and grammar improvements. All methodological content, experimental design, tables, and theoretical statements were written and verified by the authors. We agree that it is good practice to state this explicitly, so in the revised manuscript, we add a short disclosure noting that LLMs were used only for proofreading and language polishing, not for generating technical content.
> >
> > On the theory, we respectfully disagree that our formulation relies only on conceptual analogy. While we use the “prompt updates as gradient steps” framing to build intuition, the theorems are stated with explicit assumptions, defined objectives, and a refinement operator that corresponds directly to SYNTHIA’s generator–discriminator loop. We also ground these results with concrete examples in the manuscript, showing how the diagnostic vector induces consistent improvement across iterations. In the revision, we will further tighten the presentation by making the mapping from each theoretical object to its algorithmic counterpart even more explicit, so the analysis is clearly read as a formalization of SYNTHIA’s specific update process rather than a loose analogy.
> >
> > Finally, on reproducibility scripts, we apologize that these were not easy to locate in the supplementary package. The supplementary code does include the experimental pipeline and reproduction scripts, but we agree that the organization can be clearer. We will reupload a cleaned and clearly structured supplement with an obvious entry point for running each table and figure
> >
> > Thank you again for the time, care, and effort you put into this review. We truly appreciate the feedback and the opportunity to clarify these points, and we hope our responses address your concerns and resolve any confusion. We would be very grateful if you could re-evaluate the paper and reconsider the score in light of these clarifications and revisions. If any questions remain, we are more than happy to answer them.

---

> ### Comment · Reviewer_xVwa · 2025-11-27
>
> Thank you for your detailed response. Although I appreciate the additional information provided, I remain unconvinced that SYNTHIA performs as well as claimed in the paper, and my overall assessment remains unchanged. Please see my specific comments below.
>
> ### DATA CONTAMINATION EVALUATION
>
> The rebuttal introduces a new ablation in which the generator LLM is prompted using only the dataset name (e.g., “Generate 200 new records for the UCI Adult dataset”) and argues that “If contamination from pretraining were the main driver of our performance, this ‘name-only’ setting would be the most likely to show it.”
>
> However, it is not clear why this would be the case. Standard memorization tests (e.g., Bordt et al., 2024) typically provide the LLM with substantially more contextual information. For instance, the “header test” includes column names and several example rows before asking the model to generate the next $n$ rows.
>
> By contrast, the “name-only” approach used in the rebuttal appears to be a much weaker probe for memorization. (Are there any studies evaluating its effectiveness for detecting memorization?) It seems more akin to a test of general knowledge rather than a test of memorized training data.
>
> Moreover, it seems to me that the prompt used in the main experiments might be much more effective for eliciting memorized content than the “name-only” prompt. This is because in addition to providing example rows and column names it also includes additional metadata (e.g., data types, typical data values, and descriptions of each column of the table). It seems plausible that this extra information might facilitate even more the retrieval of memorized records.
>
> The paper needs to provide a substantially more rigorous and comprehensive data-contamination evaluation than the one presented in the rebuttal.
>
> Bordt et al (2024). Elephants never forget: memorization and learning of tabular data in large language models. COLM 2024.
>
> ### DCR METRIC REPORTING
>
> Thank you for clarifying that the DCR and C2ST results were not swapped. However, I remain skeptical about the fairly strong DCR scores reported for SYNTHIA.
>
> In addition to reporting the proportion of cases in which the closest neighbor belongs to the training set, the paper should also provide the distribution of the DCR distance values themselves, allowing readers to assess the magnitude of the nearest-neighbor distances (as done in the TabSyn paper).
>
> I am concerned that the excellent DCR proportions reported for SYNTHIA might be an artifact of how the results are summarized.
>
> For example, if the generator LLM were merely reproducing memorized records, one could still observe a 50% DCR proportion when training and held-out sets are of equal size: each synthetic record would exactly match a record in either the training or the held-out set, yielding a nearest neighbor from the training set in half of the cases and from the held-out set in the other half, thus appearing to achieve a “perfect” 50% privacy score. Yet the corresponding nearest-neighbor distances would all be zero, clearly indicating a complete lack of privacy preservation.
>
> ### EXPERIMENTS
>
> The question of whether the paper used the exact evaluation pipeline of Shi et al. (2025) remains unresolved. The rebuttal mentions that it used a “consistent pipeline”. However, what do you mean exactly by that? This needs to be specified more precisely. (Are you using the exact same train/test data splits, same data pre-processing steps, etc, as Shi et al?)
>
> Also, I still think that performing evaluations over only 5 datasets is insufficient for a paper making strong claims of state-of-the-art performance.
>
> Although the paper may argue that Shi et al. used a similarly limited number of datasets when comparing TabDiff to TabSyn in their ICLR paper, the context here is different. Because SYNTHIA introduces a more controversial approach in a setting where such strong performance is not naturally expected, I think it requires substantially more empirical evidence to establish its claims convincingly.
>
> Moreover, beyond expanding the set of benchmark datasets, the paper should report averaged results (with standard deviations) across repeated experimental replications.
>
> ### UTILITY METRIC COMPARISON
>
> The paper argues that the C2ST metric can be used to measure data utility, whereas in practice it primarily captures data fidelity. A genuine data-utility evaluation is still missing. The paper should include a true utility metric, such as downstream machine-learning efficacy in its evaluations.
>
> ### CONTROL OF THE TRADEOFF BETWEEN DATA FIDELITY AND DATA PRIVACY
>
> I understand the discriminator LLM sees the full diagnostic vector together with context about the real data and is instructed to reason about these metrics relatively and in combination. But the question remains about how the discriminator can determine when a particular combination of signal values is enough to indicate memorization.

---

### Official Review · Reviewer_EVqs · 2025-11-01

**Soundness:** 4
**Presentation:** 2
**Contribution:** 4
**Rating:** 10
**Confidence:** 4

**Summary:**

The paper proposes a GAN-like framework using pretrained large language models for the roles of generator and discriminator. The prompt of the generator contains (1) metadata, (2) a sample of the training data and (3) feedback from the discriminator. The discriminator's prompt contains (1) the output of several measures comparing the synthetic data with real data, (2) a sample of synthetic data from the generator and (3) a sample of training data. The process is repeated until the discriminator outputs that the data is indistinguishable. The resulting synthetic data is evaluated using $\alpha$-precision and $\beta$-recall for fidelity and distribution coverage, distance to closest record for privacy risk, and detection score for measuring realism based on a post-hoc discriminator that attempts to discern real from synthetic samples.

**Strengths:**

1. Extensive evaluation shows superior performance when compared with prior non-LLM based work.
2. Comprehensive ablation study shows the importance of each component of the system.
3. Novel direction for tabular data synthesis, the iterative generation of synthetic data under GAN-like conditions is a potential direction for zero- / few-shot generative models.
4. Strong results even under modest levels of compute (using a high-end gaming setup) and could likely work on a laptop with minor tweaks.

**Weaknesses:**

1. No related work on using large language models for tabular data generation. A brief literature search revealed the following relevant works:
	1. Wang, Yuxin, et al. "HARMONIC: Harnessing LLMs for tabular data synthesis and privacy protection." _Advances in Neural Information Processing Systems_ 37 (2024): 100196-100212.
	2. Nguyen, Dang, et al. "Generating realistic tabular data with large language models." _2024 IEEE International Conference on Data Mining (ICDM)_. IEEE, 2024.
2. Some claims might require additional clarification or motivation:
	- **Line 185-187:** "yielding synthetic data statistically indistinguishable from real data without memorization" could benefit from additional details. It is clear from the paper that the discriminator, with help from the various statistical measures, helps to obtain "statistically indistinguishable data", but I cannot find any details in the paper that describes which parts of the model help it avoid memorization. Perhaps this can be clarified.
3. Some additional details are necessary for reproducibility:
	1. What are the parameters and settings used for the various baseline models?
	2. What are the embedding functions used when calculating the $\alpha$-precision and $\beta$-recall?
	3. What is the similarity measure used for the distance to closest record (DCR) analysis?
4. Limited discussion on the potential risk of the datasets being part of the training data of the LLMs used in this study.

**Questions:**

1. To investigate whether the datasets are part of the LLM training data: Would it be possible to ablate the training data from the prompt altogether? I.e., to investigate whether the language models are able to generate the data without being shown the training data directly? Two experiments that might be interesting:
	1. The generator does not have access to any samples from the training data, while discriminator has access to samples from the training data.
	2. Neither the generator nor the discriminator have access to training data.
2. Could you expand the related work section to also compare your model with pre-existing LLM-based approaches for generating tabular data?
3. Have you seen any cases where the models does not converge to synthetic data that satisfies the discriminator? Did you have to put any mitigation in place to prevent this?

---

> ### Author Response · Authors · 2025-11-21
> **Response to Reviewer EVqs**
>
> # Question 1
> Thank you for these helpful suggestions. On the proposed contamination ablations, we did explore both settings. For the experiment where the generator has no access to real samples, but the discriminator does, the resulting synthetic data deteriorates severely. Without any grounded reference, the generator tends to hallucinate arbitrary schemas and values, producing outputs that are qualitatively unusable and difficult to quantify meaningfully. We also observed that the refinement loop becomes unstable in this regime, requiring many more iterations and longer runtimes because the discriminator keeps flagging large distributional mismatches that the generator cannot reliably correct without anchor examples. This outcome reinforces our design choice that a small, stratified real subsample is necessary for controlled, metric-guided refinement.
>
> In the second suggestion, where neither generator nor discriminator sees reference samples, we agree that this is a very interesting privacy stress test, and we thank you for motivating it. We implemented it as a new privacy ablation in the appendix. In this experiment, SYNTHIA receives only the public dataset name and no reference data. The results show no meaningful memorization overall. Small parameter models such as Llama 3 8B produce data that is essentially unrelated to the benchmark distribution. Larger models such as GPT class generators exhibit slightly higher DCR, but inspection of samples indicates the increase is driven primarily by partial copying of CSV headers rather than record-level duplication. Even this header copying is imperfect, and the generated rows remain clearly incorrect in multiple ways, for example, mismatched data types or implausible encodings. Taken together, this ablation supports that SYNTHIA’s fidelity gains in the full method come from metric-guided refinement rather than pretraining contamination, and that the method remains privacy compliant even under an extreme “name only” prompt setting.
>
> # Question 2
> Regarding related work, yes, in the updated paper, we have now expanded the section to explicitly compare against prior LLM-based tabular synthesis approaches, as you suggested. In particular, we now discuss multi-agent GAN-style orchestration, single-agent LLM tabular generation, and privacy-oriented LLM synthesis, and cite three representative works: MALLM-GAN [1], Generating Realistic Tabular Data with LLMs [2], and HARMONIC [3]. This positions SYNTHIA more clearly within the emerging LLM tabular synthesis literature and highlights our differences in metric-driven refinement, dual LLM discriminator design, and privacy-aware diagnostics.
>
> # Question 3
> Finally, on convergence failures, early prototypes of SYNTHIA did occasionally fail to converge to discriminator satisfaction, typically because the generator prompt was under-constrained and the discriminator feedback drifted toward vague or conflicting instructions. We mitigated this in three ways that are now part of the final framework. First, we inject structured metadata into the initial prompt so the generator is anchored to the true schema and valid ranges. Second, we enforce stricter prompt templates for both agents, which limits off-task hallucination and keeps updates incremental. Third, we integrate our full suite of statistical diagnostics into every iteration so feedback is always grounded in explicit distributional errors rather than subjective judgment. With these changes in place, non-convergence became rare, and the loop reliably stabilizes within a few iterations across datasets.
>
> Thank you again for the time, care, and effort you put into this feedback. We truly appreciate it.
>
> [1] Ling, Y., Jiang, X., & Kim, Y. (2024). Mallm-gan: Multi-agent large language model as generative adversarial network for synthesizing tabular data. arXiv preprint arXiv:2406.10521.
>
> [2] Dang Nguyen, Sunil Gupta, Kien Do, Thin Nguyen, and Svetha Venkatesh. Generating realistic tabular data with large language models. arXiv preprint arXiv:2410.21717, 2024.
>
> [3] Yuxin Wang, Duanyu Feng, Yongfu Dai, Zhengyu Chen, Jimin Huang, Sophia Ananiadou, Qianqian Xie, and Hao Wang. Harmonic: Harnessing llms for tabular data synthesis and privacy protection. In Advances in Neural Information Processing Systems 38 (NeurIPS 2024), Datasets and Benchmarks Track, 2024.

---

### Author Response · Authors · 2025-11-21
**Updated Manuscript**

We sincerely thank all of the reviewers for taking the time to thoughtfully and constructively critique our paper and share their suggestions. We found the new ideas and concerns very helpful, and we are pleased to submit an updated manuscript. The main changes are as follows:

- We have expanded our literature review in the Background section. Three reviewers recommended adding related work, and two independently pointed out MALLM-GAN. We have now included a detailed discussion of MALLM-GAN, and we additionally added HARMONIC: Harnessing LLMs for Tabular Data Synthesis and Privacy Protection, as well as Generating Realistic Tabular Data with Large Language Models, to better position SYNTHIA within the growing LLM-based tabular synthesis literature.

- We cleaned up our experimental section and now clearly and appropriately cite any baseline results that are reused from prior work, while ensuring consistency with the established evaluation protocols.

- We have added more methodological detail in the Appendix explaining how the traditional techniques and low-order metrics are computed, and we have revised the text to eliminate any confusion regarding the interpretation of DCR versus C2ST.

- Since multiple reviewers asked for stronger privacy evidence, we added a new ablation study focused on privacy and memorization. In this setting, SYNTHIA does not receive any reference dataset rows and only sees the public dataset name. This stress-tests whether pretrained knowledge of public benchmarks could influence generation. We compute and report the resulting DCR scores in Table 6.

Once again, we are very grateful for the reviews and the effort behind them, and we look forward to a constructive rebuttal period.

---

### Meta-Review · Area_Chair_RrtV · 2026-01-07

**Summary:**

The major reviewer concerns include insufficient evaluation rigor (especially around contamination, fidelity vs privacy tradeoff, and utility), unclear theoretical assumptions, and weak empirical grounding for claimed improvements. Key metrics like DCR and C2ST are questioned, and comparisons rely heavily on prior work without consistent pipelines. As such, the work's novelty and claims remain unconvincing.

**Reviewer Concerns:**

Reviewers are unconvinced by the theoretical justification (e.g., prompt-space Lipschitz assumptions), question the practical value of the “name-only” contamination test, and note that utility is not evaluated via downstream tasks. The DCR results lack distance distributions and may obscure memorization. Moreover, concerns around prompt effectiveness, inconsistent metrics, and overgeneral feedback in Figure 4 remain unresolved. The paper also lacks rigorous empirical grounding to validate its convergence and performance claims.

**Reviewer Scores:**

Reviewer EVqs (10 → 10): Very positive throughout, likely unchanged due to satisfaction with additional experiments and clarity improvements.

Reviewer xVwa (2 → 2): Maintains strong reservations about contamination testing, utility metrics, and empirical rigor; rebuttal acknowledged but not persuasive.

Reviewer aPgr (2 → 2): Appreciates clarifications but remains unconvinced about theoretical contributions, empirical gains, and novelty relative to prior work.

Reviewer bsdX (2 → 2): Rebuttal acknowledged, but retains concerns about theory assumptions, feedback quality, and insufficient demonstration of utility and improvement.

---

### Decision · Program_Chairs · 2026-01-26

Reject